# 3D atomic-scale metrology of strain relaxation and roughness in Gate-All-Around transistors via electron ptychography

Shake Karapetyan [1], Steven E. Zeltmann [1,2], Glen Wilk[3], Ta-Kun Chen[4], Vincent D.-H. Hou[4] & David A. Muller [1,5] ✉

Next-generation semiconductor devices are adopting three-dimensional (3D) architectures with feature sizes in the few-nanometer regime, creating a need for atomic-scale metrology to identify and resolve performance-limiting fabrication challenges. X-ray methods provide 3D information but lack atomic resolution, while conventional electron microscopy offers limited depth sensitivity. Here we show how multislice electron ptychography, a computational microscopy technique with sub-Ångström lateral and nanometer-scale depth resolution, enables 3D imaging of buried device structures. We image prototype gate-all-around transistors and directly quantify roughness, strain, and defects at the interface of the 3D gate oxide wrapped around the channel. We find that silicon in the 5-nm-thick channel relaxes away from the interfaces, leaving only ~60% of atoms in a bulk-like structure. From a single dataset, ptychography provides quantitative metrology of atomic-scale interface roughness in 3D, previously accessible only through indirect inference, along with strain and other structural parameters needed for device modeling and process development.

The increasing demands of modern computing and the drive to sustain Moore's Law have led the semiconductor industry to transition from planar to three-dimensional transistor architectures, such as gate-all-around (GAA) structures. This shift enables improved transistor density, performance, and power efficiency, critical for advanced integrated circuits[1]. GAA transistors, with their nanosheet-based design (Fig. 1a–d), provide enhanced electrostatic control by completely surrounding the channel with the gate electrode, unlike planar field-effect transistors (FETs) or FinFETs, which lack full gate-channel coupling[1-5]. This design is ideal for ultrascaled nodes but, as shown in Fig. 1d, the intricate atomic-scale features—including interfaces, potential defects, and combinations of crystalline and amorphous materials of various composition—combined at such small length scales present significant challenges for structural and materials characterization.

Various imaging techniques have been employed to characterize semiconductor devices over the decades, most of which provide only two-dimensional information, whereas 3D metrology tools are necessary to fully characterize the atomic-scale structure and complexity of modern three-dimensional device architectures. Scanning transmission electron microscopy (STEM) with electron energy loss spectroscopy, for instance, was used to establish the critical thickness of amorphous silicon oxide required for bulk-like electronic properties[6]. X-ray ptychographic tomography provides 3D imaging at deep submicron scales, offering insights into larger features like interconnects and the very intricate back-end elements[7]. Atom probe tomography (APT) enables nm-scale characterization with chemical specificity, though challenges exist due to local charging effects near insulators[8]. Atomic electron tomography (AET) has reported atomic-scale 3D

[1]School of Applied and Engineering Physics, Cornell University, Ithaca, NY, USA. [2]Platform for the Accelerated Realization, Analysis, and Discovery of Interface Materials (PARADIM), Cornell University, Ithaca, NY, USA. [3]Advanced Semiconductor Materials (ASM) America, Phoenix, AZ, USA. [4]Corporate Analytical Laboratories, Taiwan Semiconductor Manufacturing Company, Hsinchu, Taiwan. [5]Kavli Institute at Cornell for Nanoscale Science, Cornell University, Ithaca, NY, USA. ✉e-mail: david.a.muller@cornell.edu

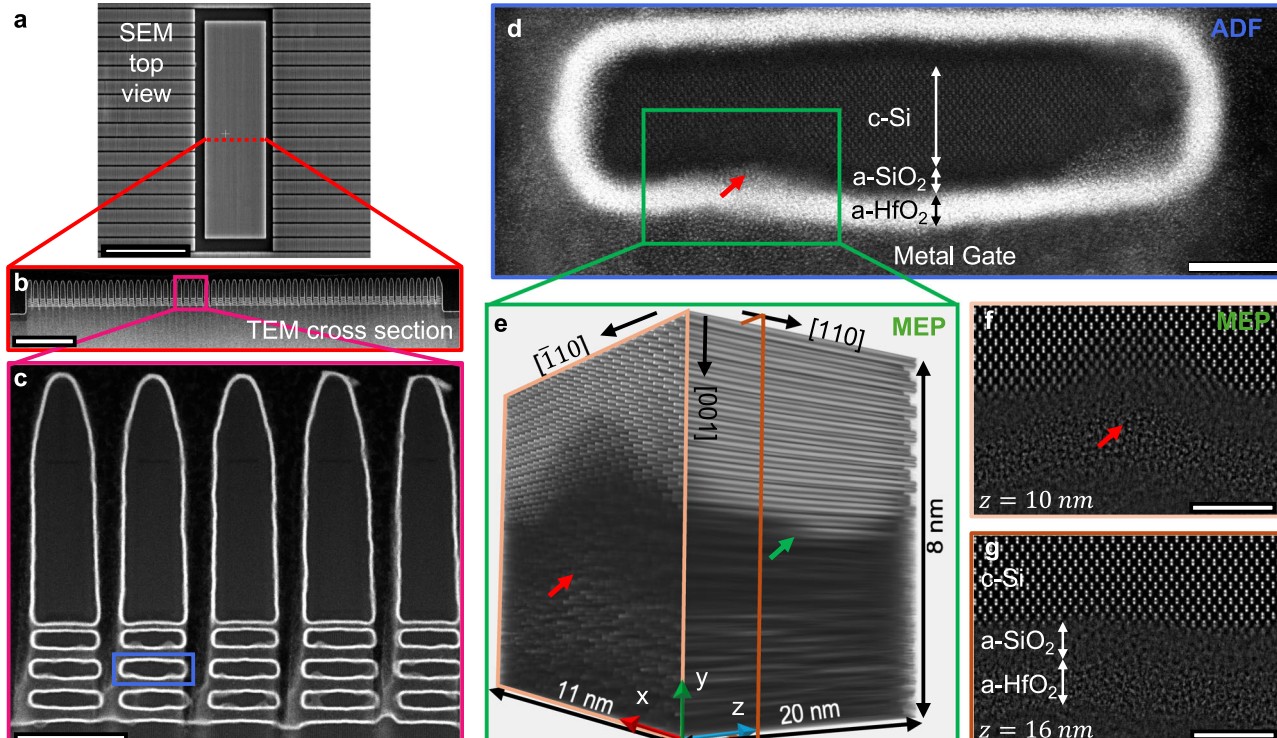

**Fig. 1 | Overview of the gate-all-around structure spanning details from microns to Ångströms with depth resolved imaging enabled by multislice electron ptychography. a** SEM top view of an imec gate-all-around test structure node (early process development, expected to contain many defects) with a red dotted line indicating the region where a depth-wise electron-transparent cross-section (lamella) was prepared (scale bar: 3 µm). **b** ADF image of the indicated cross-section showing 63 gate-all-around columns (three per stack; scale bar: 0.5 µm). **c** Magnified view of the pink square in **b**, highlighting five gate-all-around columns; one structure is highlighted in blue and shown in **d** (scale bar: 50 nm). **d** ADF image of a single gate-all-around structure with labeled components (crystalline-silicon (c-Si)/amorphous-silicon oxide (a-SiO₂)/amorphous-hafnium oxide (a-HfO₂)); thin black and white arrows indicate the extents of the Si channel and surrounding amorphous layers). Hafnium oxide intrusions into the silicon channel are visible, one of them highlighted with a red arrow. The green-marked region was imaged by MEP (results shown in panels **e–g**) (scale bar: 5 nm). **e** 3D MEP reconstruction showing interface roughness, a hafnium oxide intrusion (red arrow) and a step edge at the crystalline-silicon/amorphous-silicon oxide interface (green arrow). These buried features, important for device performance, while readily apparent with MEP, can be inaccessible or easily missed with conventional imaging methods. **f** MEP depth slice at z = 10 nm from the region in **e** (light orange plane) showing both light (Si, O) and heavy (Hf) atoms, visible due to the linear atomic-number contrast (scale bar: 3 nm). **g** MEP depth slice at z = 16 nm from the region in **e** (dark orange plane), showing that the hafnium oxide intrusion (red arrow in **f**) is localized in depth and does not extend through the entire interface (scale bar: 3 nm).

imaging in small nanoparticles, particularly for heavier elements[9]. However, its reliance on annular dark field (ADF) STEM limits sensitivity to light elements and introduces channeling artifacts in thicker samples, making it challenging to scale to larger, device-relevant fields of view. Conventional atomic-resolution imaging techniques, while capable of sub-Ångström lateral resolution, often fail to provide depth-resolved structural information and suffer from well-documented violations of the projection approximation from multiple scattering artifacts[10–12].

As the critical dimensions of the GAA transistors approach the sub-10 nm regime, the limitations of existing metrologies become increasingly pronounced. Even a single defect or pinhole (a small void or discontinuity at the dielectric-semiconductor interface) can significantly impact device performance, yet such features remain difficult to detect with current techniques[1,13–15]. Interface roughness and strain are particularly important to quantify, as they can strongly degrade carrier mobility and shift threshold voltage[14–21]. Indeed, recent quantum transport simulations confirm surface-roughness scattering as the predominant factor limiting electron mobility in ultrascaled silicon devices[21]. However, there is a large variance in previously reported measurements of interface roughness[21–24], indicating both sensitivity to processing conditions and the limitations of conventional projective imaging methods, which tend to underestimate the true three-dimensional nature of interface roughness. These challenges underscore the need for direct, atomic-resolution 3D measurements to more accurately characterize buried interfaces and enable predictive models of device behavior. This need has been explicitly identified in recent semiconductor roadmaps and national initiatives, including the CHIPS for America Act's Grand Challenges and review articles[1,25].

Here, we show how multislice electron ptychography (MEP) addresses the critical metrology gap in device characterization highlighted above, namely, the need for Ångström-scale resolution, three-dimensional imaging capabilities, and simultaneous sensitivity to both light and heavy atoms[26–28] (in the context of GAA devices, light elements refer to Si and O, while heavy refers to Hf), thereby enabling direct measurement of atomic structure, strain, and interface roughness in 3D. By scanning a convergent electron probe across the sample and collecting diffraction patterns from overlapping illuminated regions, MEP retrieves the phase shift introduced by the object's atomically-resolved electrostatic potential, providing nanometer-scale depth information through post-processing of scanning diffraction data[29]. We demonstrate that MEP enables high-fidelity three-dimensional imaging of GAA transistors, validate its accuracy on simulated structures and show experimentally that it outperforms conventional through-focal STEM in resolution and dose efficiency. Using MEP on early-stage GAA test structures from Interuniversity Microelectronics Center (imec), we directly image stacking faults, pinholes, interface roughness, and strain relaxation in the crystalline-Si (c-Si) channel,

capturing buried detrimental structural features well before electrical testing is possible. In these GAA devices, in the c-Si channel (labeled c-Si in Fig. 1d), we measure the structural transition from strained interfacial silicon to bulk-like silicon to span over 2 nanometers of the 5-nanometer-wide channel (over 40%). Comparing these findings to a conformal amorphous-$SiO_2$ (a-$SiO_2$) layer on epitaxial c-Si, we find that the strain relaxation is process-dependent and can serve as a metric for interface quality. By tracking interfacial atoms, we extract 3D maps of interface roughness amplitude and correlation length, uncovering asymmetries tied to the processing history of the interfaces. Formation of the critical GAA structure occurs early in the overall CMOS fabrication process, which comprises roughly a thousand steps and typically takes 3-4 months to complete from start to finish. Early structural feedback from MEP performed on dedicated witness wafers (test wafers fabricated alongside production wafers for process monitoring, including by destructive analysis techniques) can provide rapid insight into structural defects and interface quality early in the fabrication flow. This will accelerate the learning cycle during process development by enabling direct correlation between process changes, structural outcomes, and complementary short-loop electrical testing, reducing costly iterations. Moreover, since MEP captures atomic-scale interface roughness and strain, which are both key determinants of carrier mobility, it offers a direct structural metric for building predictive models of device performance and yield, enabling earlier-stage screening and optimization.

## Results

### Three-dimensional imaging by depth sectioning

Accurate atomic-scale measurements of interface roughness and structural transitions in 3D are essential for understanding and modeling the electronic behavior of advanced semiconductor devices[16–20]. While (S)TEM analysis is inherently destructive, requiring samples to be prepared from a chip portion (Fig. 1a, b), it is standard practice in industry to dedicate test structures on a witness wafer for this analysis. Figure 2 provides an overview of the microscope configuration (Fig. 2a), the depth-information acquisition principles for MEP (Fig. 2b, c) and through-focal imaging approaches (Fig. 2c, d), and a direct experimental comparison of these methods (Fig. 2e–g).

Through-focal annular dark field (tf-ADF) and through-focal integrated differential phase contrast (tf-iDPC) are conventional STEM imaging methods that can potentially obtain limited 3D information by acquiring a series of 2D images, each with a small depth of field, at different defocus values (Fig. 2c, d)[10,12,30–35]. tf-ADF uses high-angle scattering (Fig. 2a, blue), which, in very thin samples, exhibits approximately $Z^2$-dependent contrast (Fig. 2f). tf-iDPC uses low-angle scattering (Fig. 2a, purple) to approximate the projected potential, yielding, in very thin samples, linear or sublinear atomic number contrast and improved sensitivity to light elements (Fig. 2g)[36].

Both methods rely on a linear, weak-phase imaging approximation—i.e., that the probe shape remains unchanged from its free-space solution as it propagates through the specimen—an assumption that fails for device-relevant thicknesses (tens of nanometers) due to multiple scattering. At these thicknesses, the probe wavefront is repeatedly altered as it interacts with successive atomic planes (multiple scattering) and can be guided along the positively-charged columns of atomic nuclei (channeling)[37], breaking the direct relationship between image intensity and the underlying atomic potential. These nonlinear probe-sample interactions produce well-documented artifacts[10,11,30,36], including axial elongation, contrast reversals, and positional inaccuracies, which lead to mis-localized interfaces and spurious depth-dependent features (Fig. 2e–g).

MEP uses the same optical configuration as tf-ADF and tf-iDPC (Fig. 2a) but overcomes their limitations—probe-sample coupling and the lack of any correction for multiple scattering in thicker specimens—by recording a momentum-resolved scattering distribution and reconstructing both the probe and the sample potential[38,39]. The multiple scattering is accounted for using a multislice forward model[27,40,41] (Fig. 2b). In practice, this is achieved by acquiring a four-dimensional scanning transmission electron microscopy (4D-STEM) dataset: a defocused probe is raster-scanned over the sample, ensuring overlap between adjacent illuminated regions, and recording a two-dimensional diffraction pattern at each probe position in real space. The MEP reconstruction from these datasets explicitly accounts for the probe shape and its evolution—including multiple scattering, channeling, partial coherence, and sample tilts—by iteratively propagating the recovered probe though successive slices of the specimen (Fig. 2c), enabling quantitatively accurate 3D reconstructions even for ~40 nm-thick specimens.

In MEP, the depth information is encoded via parallax: each Ronchigram acts as a shadow image of the sample with the observer effectively located at the probe crossover point. With the probe crossover ~10 nm above the sample, atoms at different depths shift across the defocused diffraction patterns at different rates as the probe scans (by parallax), allowing depth resolution from a single 4D dataset[42]. Unlike through-focal methods (or tomography approaches), which require multiple separate x-y raster scans, each representing a separate exposure of the sample to the electron beam, only a single x-y scan and its resulting 4D-STEM dataset is needed for MEP, from which 3D structural information is recovered (Fig. 2b–d). Thus, depth information is encoded within the same x-y raster scan (Fig. 2e), without requiring additional scans, thereby reducing the total electron dose.

The resolution of tf-ADF and tf-iDPC is bounded by the diffraction limit set by the numerical aperture of the lenses. The lateral Rayleigh resolution and depth resolution are given by

$$d_{xy} = \frac{0.61\lambda}{\alpha} \tag{1}$$

$$d_z = \frac{2\lambda}{\alpha^2} \tag{2}$$

where $\lambda$ is the electron wavelength and $\alpha$ is the convergence semi-angle[43]. For a modern aberration-corrected instrument operated at typical settings of 300 kV ($\lambda = 1.97$ pm) with $\alpha = 30$ mrad, these correspond to lateral and depth resolutions of 0.40 Å and 44 Å, respectively. Increasing $\alpha$ further is limited by chromatic aberrations and longitudinal source coherence which degrade image quality[43,44]. These Rayleigh limits are upper bounds; multiple scattering in real samples degrades resolution beyond the ideal values.

Recent high-speed, high-dynamical-range hybrid-pixel detectors[45,46] now enable routine collection of high-quality 4D-STEM datasets suitable for MEP. By capturing scattering to high angles ($\beta$, Fig. 2a), these detectors extend the accessible spatial frequencies, enabling lateral and depth resolution beyond the diffraction limit arising from the limited probe-forming-aperture semi-angle, $\alpha$. Under high-dose conditions, the ultimate resolution in MEP is therefore bounded by the detector's maximum collection angle $\beta$ ($> \alpha$) rather than $\alpha$ alone (Fig. 2a)[27,47–49].

This combination of dose efficiency, resolution, and depth sectioning has enabled 3D imaging across a wide range of materials using MEP[28,42,50–56], including our initial conference abstracts on the crystalline-Si/amorphous-$SiO_2$ interface[57–59]. The acquisition process is no more complex than that of a standard STEM image, and with recent algorithmic advances[60], initial reconstructions that once took a day can now be completed in under an hour (see Typical MEP Workflow Duration in Methods).

### Benchmarking of the imaging techniques

To systematically assess imaging accuracy, we benchmarked MEP against conventional through-focal methods—tf-iDPC and

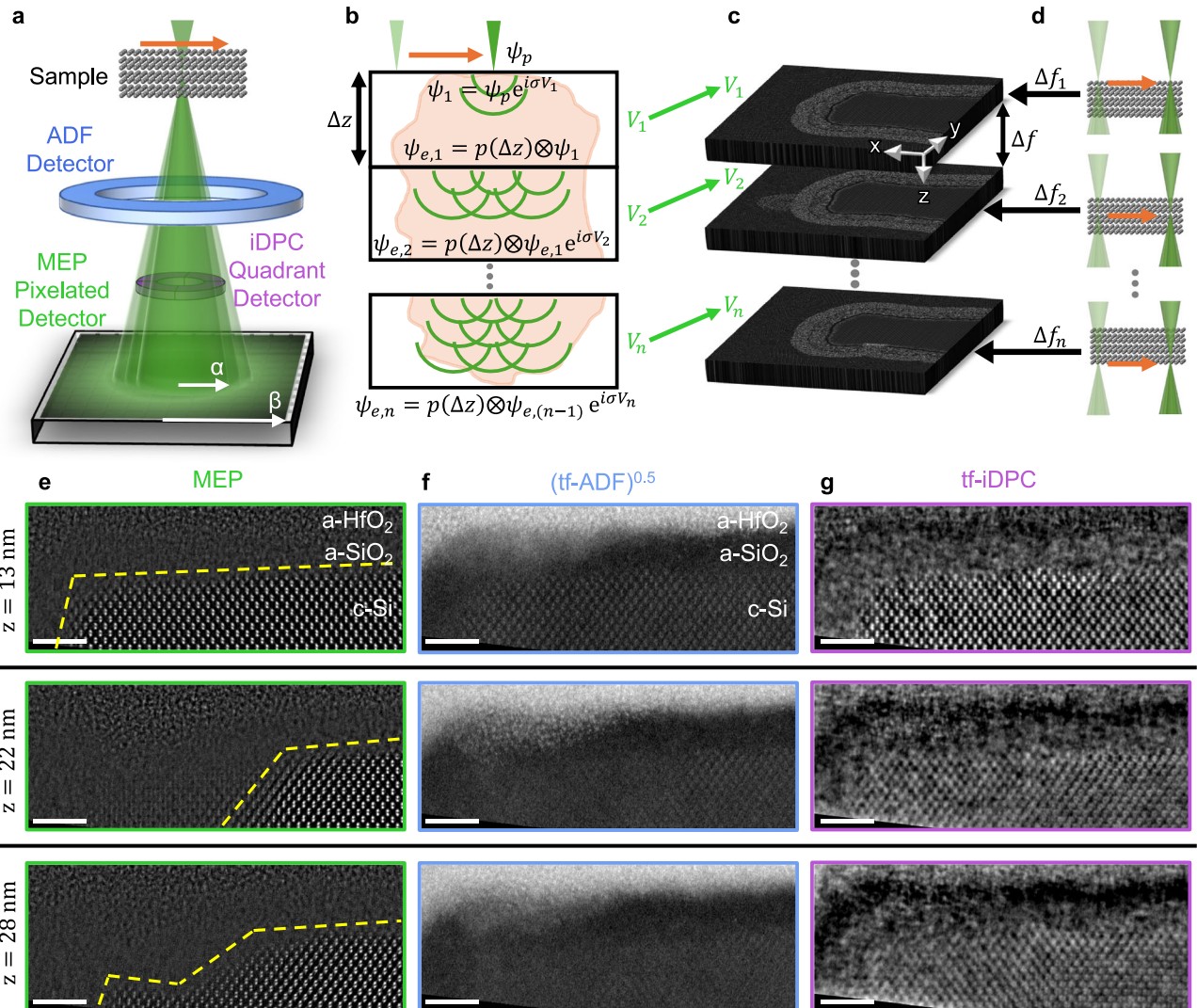

**Fig. 2 | Three dimensional imaging by depth sectioning using multislice electron ptychography and through focal imaging approaches. a** STEM geometry illustrating the converged electron probe (green, semi-angle α) raster-scanned (orange arrow) over an electron-transparent sample. Transmitted electrons are recorded using different detectors to generate distinct imaging modes: a high-angle annular dark-field (ADF) detector (blue), a quadrant detector for iDPC (purple), and a pixelated detector (black) for multislice electron ptychography (MEP) that records scattered electrons up to the detector's outer collection angle β. **b** Schematic of the MEP reconstruction, where the probe is untangled from the sample potential to recover a stack of potential slices $V_1 \dots V_n$. **c** Conceptual illustration of depth-resolved slices reconstructed by different approaches, shown here using an atomic model of a GAA device. Green arrows from panel **b** and black arrows from panel **d** indicate that such depth resolved information can, in principle, be obtained from either MEP (single dataset) or through-focal (n scans of tf-ADF/tf-iDPC) data. d Through-focal imaging scheme for tf-iDPC and tf-ADF, where a series

of 2D images is recorded at different defocus values $f_n$ and aligned to form a 3D representation of the sample convolved with the probe's point-spread function. Unlike MEP, these methods assume a weakly interacting probe and therefore neglect probe evolution from multiple scattering, leading to characteristic artifacts. **e–g** Depth-specific experimental comparison for the same GAA device region (all scale bars: 2 nm). **e** MEP reconstructed electrostatic potentials show an intact channel at 13 nm depth, a partially missing channel at 22 nm, and a distorted channel shape at 28 nm (yellow dashed outlines). Total dose: $0.5 \times 10^5$ e⁻ Å⁻². f tf-ADF images (square-root contrast shown) at corresponding defocus values obscure Si lattice contrast due to dose limitations and fail to recover the correct c-Si channel shape in depth. Total dose: $1 \times 10^5$ e⁻ Å⁻². **g** tf-iDPC images show the channel appearing intact at all depths, with poorly resolved lattice details and spurious depth information, missing the channel deformation. Total dose: $1 \times 10^5$ e⁻ Å⁻². Additional defect comparisons and depth slices are provided in Supplementary Fig. 4.

tf-ADF—using both simulations and experimental reconstructions. Figure 3 summarizes the simulated 3D benchmark of a rough crystalline-silicon (c-Si)/amorphous-silicon oxide (a-SiO₂)/amorphous-hafnium oxide (a-HfO₂) interface, evaluating how each method recovers surface boundaries, depth positions, and buried amorphous features under realistic imaging conditions. These multislice simulations (calculated using abTEM[61], see Methods) show that MEP consistently resolves crystalline boundaries, interface roughness, and buried amorphous speckle (Fig. 3a, b and Supplementary Fig. 1a, b). In contrast, tf-iDPC and tf-ADF display elongation, noise, and depth misattributions that obscure these features (Fig. 3c, d,

Supplementary Fig. 1c–d), consistent with their inability to correct for probe modification from multiple scattering and channeling[10,12]. This difference is quantified in Fig. 3e: for a ground-truth 66-Å-deep Si column, MEP retrieves the correct thickness, whereas tf-ADF and tf-iDPC overestimate it (94 Å and 93 Å) due to axial elongation and apparent surface shifts—artifacts that worsen with increased sample thickness or slight tilt.

To estimate depth blurring, we fitted error functions to the entry and exit edges of the axial line profiles (see Fig. 2e and Supplementary Fig. 2). MEP shows an edge blur of only 22 Å, while infinite dose tf-ADF shows 40 Å. The finite-dose tf-ADF profile is too noisy to fit, mirroring

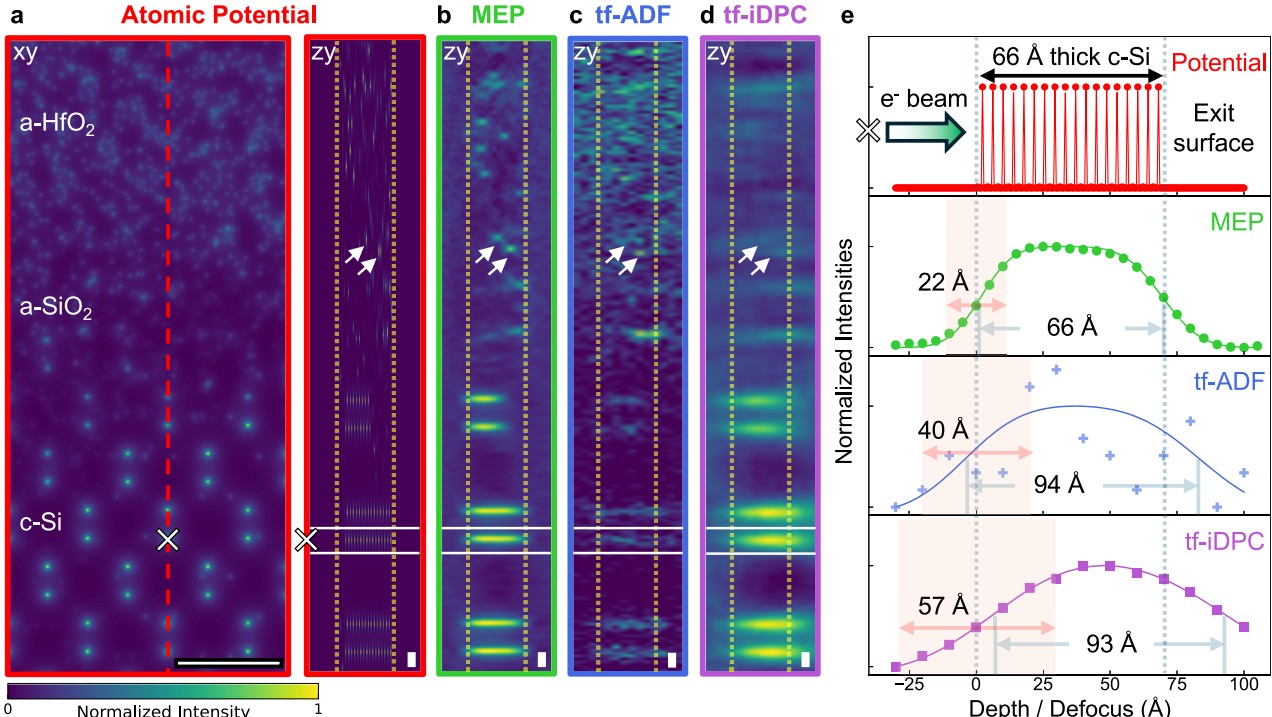

**Fig. 3 | Theoretical benchmarking of depth sectioning performance of multislice electron ptychography and through focal imaging approaches using a rough interface model. a** Projected atomic potential of the model structure (xy, scale bar: 5 Å) and ground-truth depth section along the red dashed line (zy, scale bar: 10 Å), showing the crystalline Si substrate, amorphous SiO$_2$, and amorphous HfO$_2$. The white '×' marks a representative c-Si column and the two white arrows indicate embedded a-HfO$_2$ particles. Vertical dashed lines mark the entrance and exit surfaces of the model used for comparison across all methods. Simulated depth reconstructions along the same line using MEP (**b**), tf-ADF (**c**), and tf-iDPC (**d**), including realistic geometric and chromatic aberrations and a total dose of $2.5 \times 10^5$ e$^-$ Å$^{-2}$. Scale bars: 10 Å. The relative depth positions of the interface features and the entrance and exit surfaces are visibly distinct in **b**, while **c** shows substantial noise and **d** shows axial elongation and shifts in the apparent surface locations. **e** Depth-

intensity profiles of the ×-marked c-Si column compared with the true axial extent of the atomic potential (red) corresponding to the 66 Å-thick Si segment. Gray dashed lines indicate the true entrance and exit surfaces, while the solid gray lines indicate the apparent surface locations determined from the 50% intensity crossings of the interpolated depth profiles, and their corresponding column widths. Red shaded regions indicate the edge blur extracted from asymmetric double-error-function fits. Because the noisy tf-ADF signal cannot be reliably fitted (blue crosses), blur values are extracted from the noise-free tf-ADF signal (blue line). MEP yields the smallest edge blur (22 Å), while tf-ADF (40 Å) and tf-iDPC (57 Å) exhibit substantially larger axial broadening, consistent with their reduced ability to localize features in depth. Supplementary Fig. 1 provides selected depth slices comparing the three imaging techniques. All images share a normalized intensity range from 0 to 1, scaled by the maximum value of their respective 3D stack.

the experimental situation. tf-iDPC shows a 57 Å edge blur, comparable to the full sample thickness, indicating that the column boundaries cannot be meaningfully localized. Supplementary Fig. 2 shows that MEP's depth-sectioning performance remains stable across depth-slice spacings and detector collection angles, provided Nyquist sampling is met, and sufficient intensity is captured at the highest collected scattering angles.

The improved depth fidelity of MEP translates directly to the ability to recover buried structures: using a simulation study of a pMOS device model[62], Supplementary Fig. 3 shows that MEP resolves an amorphous Ta-filled pinhole at 14 nm at the correct depth, whereas the feature is indistinct in tf-iDPC and barely visible in tf-ADF (with the silicon channel completely obscured), despite the latter using more than twice the dose.

We next applied the same evaluation framework to experimental datasets acquired on a prototype imec GAA device (Fig. 1), using all three methods under identical imaging conditions (described in Methods). A direct comparison is shown in Fig. 2e–g. Supplementary Fig. 4 highlights the improved resolution and the higher signal-to-noise ratio (SNR) achieved by MEP. MEP achieves a lateral information limit of 0.49 Å, compared to 0.66 Å for tf-iDPC and 0.83 Å for tf-ADF (Supplementary Fig. 4), using only half the electron dose ($0.5 \times 10^5$ e$^-$ Å$^{-2}$ for the single MEP dataset versus $1 \times 10^5$ e$^-$ Å$^{-2}$ for either the full tf-iDPC or tf-ADF image series).

From isolated single-atom features in the experimental MEP reconstructions (Fig. 4a, b), we measured an axial blur of $40 \pm 7$ Å (Fig. 4c and Supplementary Fig. 5). Equivalent estimates were not possible for tf-ADF or tf-iDPC because finite-dose images are too noisy (consistent with limitations seen in simulations) and individual atoms likely move between defocus frames. Thus, MEP provides sufficient detail for quantitative 3D analysis, while tf-iDPC and tf-ADF could not reliably recover depth information under comparable conditions (Supplementary Table 1).

Two factors determine the practical resolution. First, the geometric limit is set by α for tf-ADF and tf-iDPC, and by the detector's outer collection angle β for MEP. In our experiments, our choice of β was constrained by the ~40 nm specimen thickness and the detector's finite pixel count, which limited the attainable collection angle without undersampling. Second, dose can also limit the achievable resolution: because each method accesses a different fraction of the scattered electrons (Fig. 2a), not all have enough signal to reach their geometric limits. MEP collects nearly all scattered electrons (~99%), whereas tf-ADF uses <1%, and tf-iDPC—despite the high SNR in the bright-field disk—cannot convert this advantage into reliable depth localization due to uncorrected multiple scattering. As shown in Fig. 2g, tf-iDPC's axial profile is broadened to nearly the full sample thickness, reflecting its practical limitation. Additionally, the experimental MEP depth resolution is broader because a stronger regularization (0.5 vs. 0.1 in

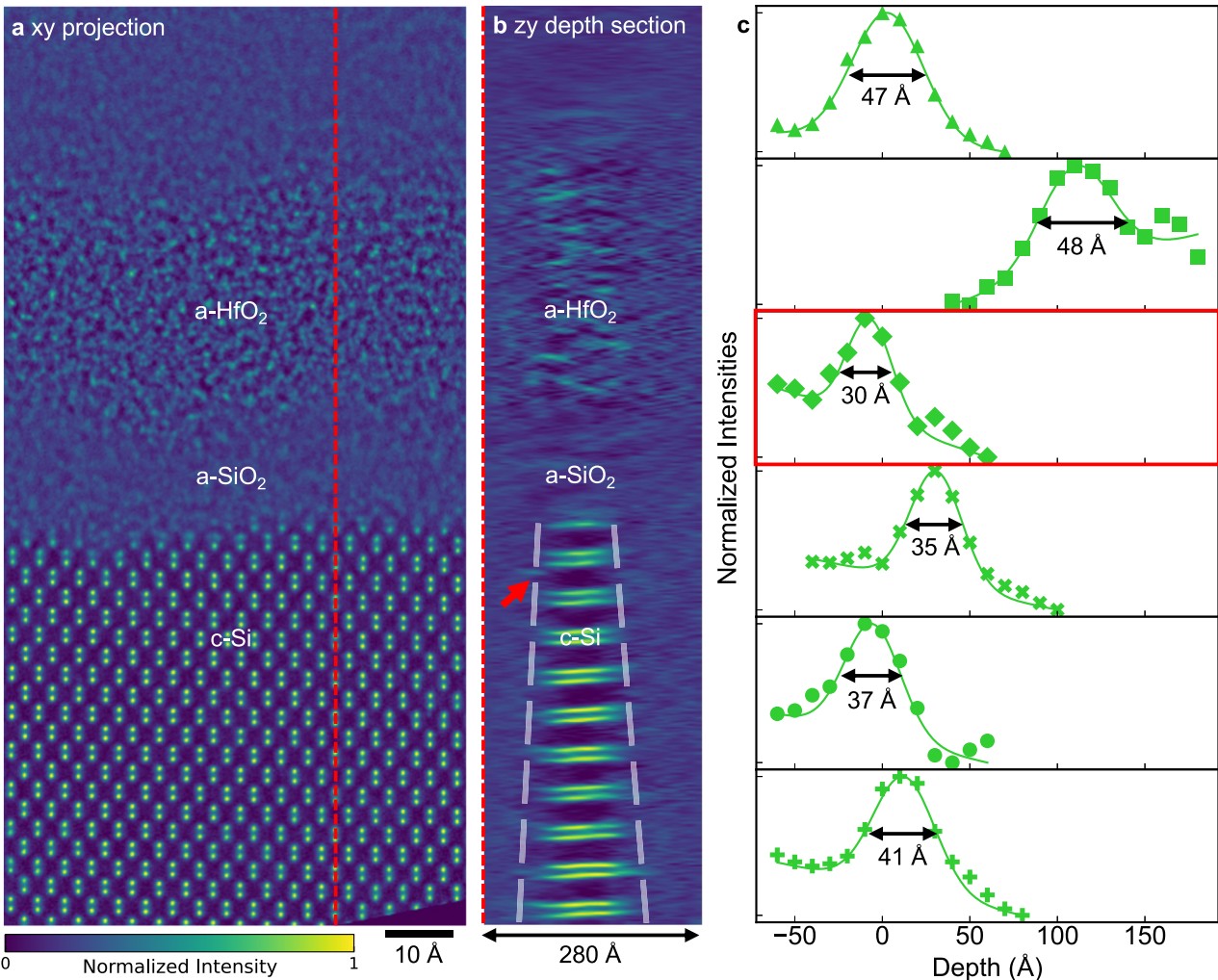

**Fig. 4 | Experimental MEP depth sections from a planar c-Si/a-SiO₂/a-HfO₂ interface and corresponding depth blur estimation. a** Projected xy view of the MEP-reconstructed electrostatic potential of a planar c-Si/a-SiO₂/a-HfO₂ interface; the red dashed line marks the location of the depth slice in panel **b** Scale bar: 10 Å. **b** MEP depth section (with entrance surface on the left, and exit surface on the right) shows internal features of the amorphous HfO₂ layer, including nanoscale contrast variations consistent with short-range order or local density fluctuations. In contrast, the a-SiO₂ portion appears more homogeneous, consistent with its lower atomic number and more uniform amorphous structure. The characteristic triangular wedge geometry of the FIB-prepared TEM lamella is visible (outlined by dashed guidelines). A single atom located at the top surface (red arrow) is identified as redeposited from the TEM sample preparation. These observations demonstrate the sensitivity of MEP to both intrinsic and preparation-induced structural features, highlighting its utility for detailed 3D characterization of amorphous-crystalline interfaces. Note that the voxel size is anisotropic in the depth direction. **c** Local axial intensity profiles along isolated single atoms from this reconstruction (see Supplementary Fig. 5 for slice locations; the red arrow in panel **b** corresponds to the third profile shown, boxed in red) together with Gaussian-plus-linear-background fits. The resulting full-width-at-half-maximum (FWHM) values are listed below each profile. Across all six atoms, the fits yield depth blurs of 30–48 Å, corresponding to an average depth blur of $40 \pm 7$ Å (s.d.). Equivalent measurements cannot be performed for tf-ADF or tf-iDPC because the finite-dose images were too noisy and individual atoms often shifted between consecutive frames. All images share a normalized intensity range from 0 to 1, scaled by the maximum value of the reconstructed 3D volume.

simulation) was used for stable convergence, which inherently suppresses high-frequency detail.

The limited depth resolution of the conventional methods also leads to ambiguity between true device features and surface damage from sample preparation. For example, a single ADF projection suggests a hafnium oxide intrusion into the Si channel (Fig. 1d) but cannot determine its depth. MEP resolves this ambiguity: its 3D reconstruction (Fig. 1e–g) shows the intrusion 10 nm beneath the lamella surface, confirming it occurred during device fabrication rather than as a TEM-preparation-induced redeposition. Additional depth sections from a planar c-Si/a-SiO₂/a-HfO₂ stack (Fig. 4a, b; Supplementary Fig. 5) likewise demonstrate that MEP distinguishes intrinsic amorphous/crystalline structure from Focused Ion Beam (FIB) preparation-related surface artifacts. The triangular wedge geometry of the lamella is clearly visible in MEP depth slices (Fig. 4b), underscoring MEP's ability to map both sample features and preparation-induced morphology.

Through simulations and experiments, we found that MEP provides quantitatively reliable 3D reconstructions with minimal depth broadening, while conventional through-focal methods are fundamentally limited by probe-evolution artifacts. Leveraging MEP's robust depth fidelity, we next quantify atomic-scale roughness and strain in GAA structures, capturing buried irregularities that directly impact device performance.

### Atomic-scale strain and roughness in gate-all-around devices

Building on MEP's depth-resolving capabilities, we now apply it to visualize and measure strain and atomic-scale roughness in gate-all-around (GAA) transistors—features that are critical for device performance[13,16,17,21] yet poorly captured by conventional methods[1,25]. Figure 5a shows reconstructed slices through a prototype GAA device, revealing sharp variations in channel structure at different depths. At 18 nm, the crystalline silicon channel appears largely intact, but at 8 nm

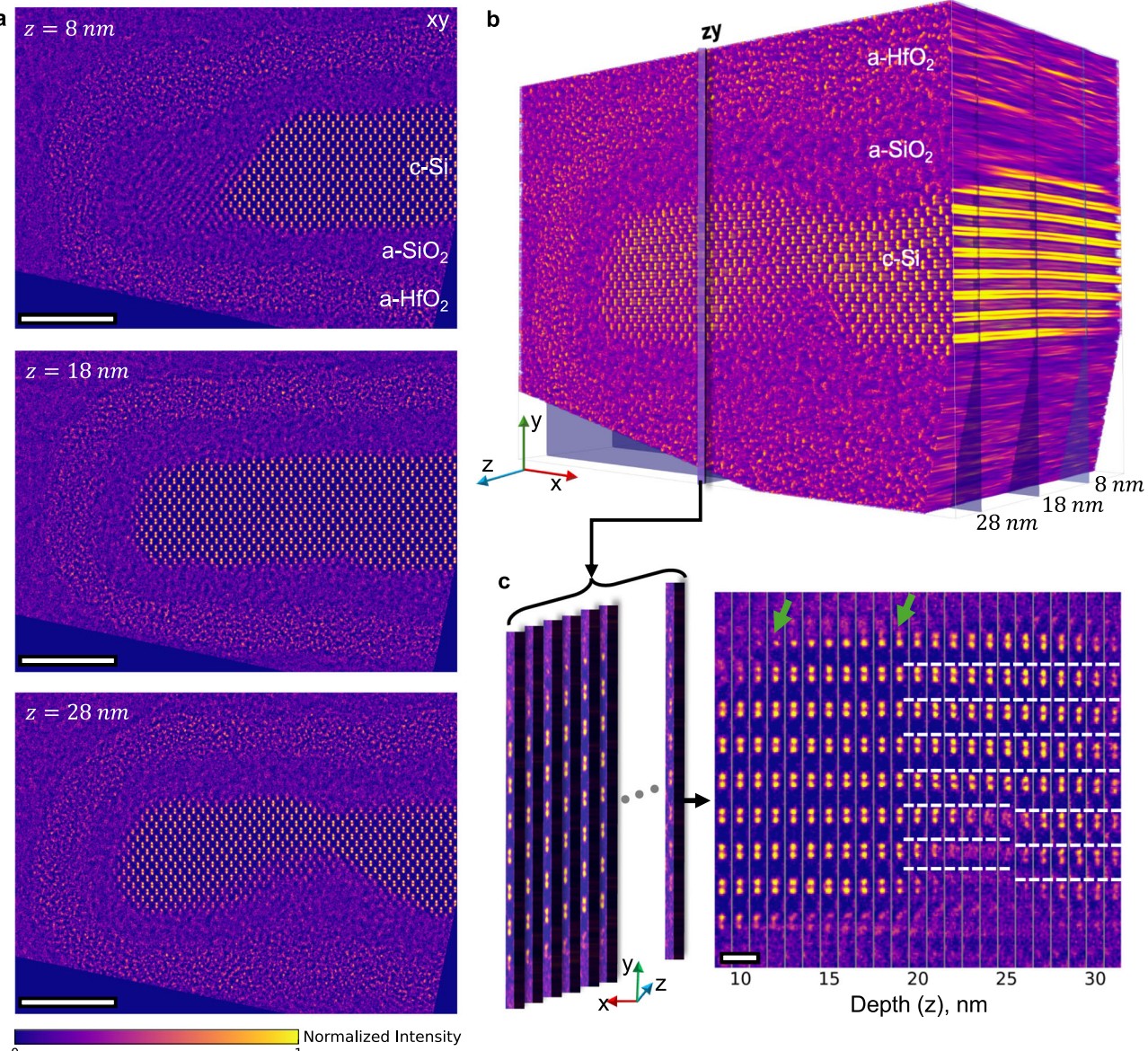

**Fig. 5 | Buried defects and rough interfaces in GAA structure 1 (GAA-1) identified using MEP. a** MEP reconstruction of GAA-1 at different depths reveals an intact channel at z = 18 nm (near the center) but irregularities at 8 nm and 28 nm, where sections of the crystalline channel are missing. Scale bars: 4 nm. **b** 3D cuboid representation of the GAA-1 reconstruction, illustrating all sliced planes with labeled components visible in three dimensions. The cuboid dimensions are 18 × 13 × 33 nm (not to scale in depth). **c** Schematic of the depth section series along the silicon column highlighted with the zy plane in **b** and a montage of those slices (scale bar: 5 Å). Step edges at the c-Si/a-SiO₂ interface are marked with green arrows. The region with missing crystalline Si is accompanied by a stacking fault, outlined with white dotted lines that have a mismatch at the stacking fault. Additional depth slices showing more stacking faults are shown in Supplementary Fig. 6. The Si channel shape and interface roughness of this structure are detailed in Supplementary Fig. 8d–f. All images share a normalized intensity range from 0 to 1, scaled by the maximum value of the reconstructed 3D volume.

and 28 nm, large sections are absent—visible as missing-channel regions ('mouse-bites')—with associated pinholes and hafnium-rich regions. The full 3D volume (Fig. 5b) places these depth slices in context, while a montage of a selected *xy* section in depth (Fig. 5c) shows a stacking fault defect, evidenced by a half-unit-cell lattice shift and bending of adjacent atomic columns. Additional views are shown in Supplementary Fig. 6. Green arrows highlight step edges at the c-Si/a-SiO₂ interface, corresponding to interface roughness captured in 3D. These early-stage GAA structures were still undergoing process development, so the presence of such structural variability was expected.

To quantitatively map interface morphology and strain, we tracked the atoms in the silicon channel across depth using Atomap[63]

(see Methods, Supplementary Fig. 7 and Movies 1–3). By grouping atoms into bilayers from the c-Si/a-SiO₂ interface inwards (Fig. 6a), we extracted Si–Si spacings per bilayer (Fig. 6b, c). We find that in these GAA structures (blue and gray), on average, Si–Si spacing relaxes to its bulk value over four bilayers or 11 Å from each interface. To put this into context, we compare this strain relaxation length to that of a planar and conformal a-SiO₂ layer on epitaxial c-Si (red), where the convergence occurs more abruptly—within just three atomic bilayers or 8 Å. Larger standard deviations in the Si–Si distances per bilayer in GAA structures (blue and gray shaded regions in Fig. 6c) compared to the planar interface (red) reflect a more strained and disordered interface, highlighting greater structural disorder compared to the planar interface. Given that in modern devices, including the ones

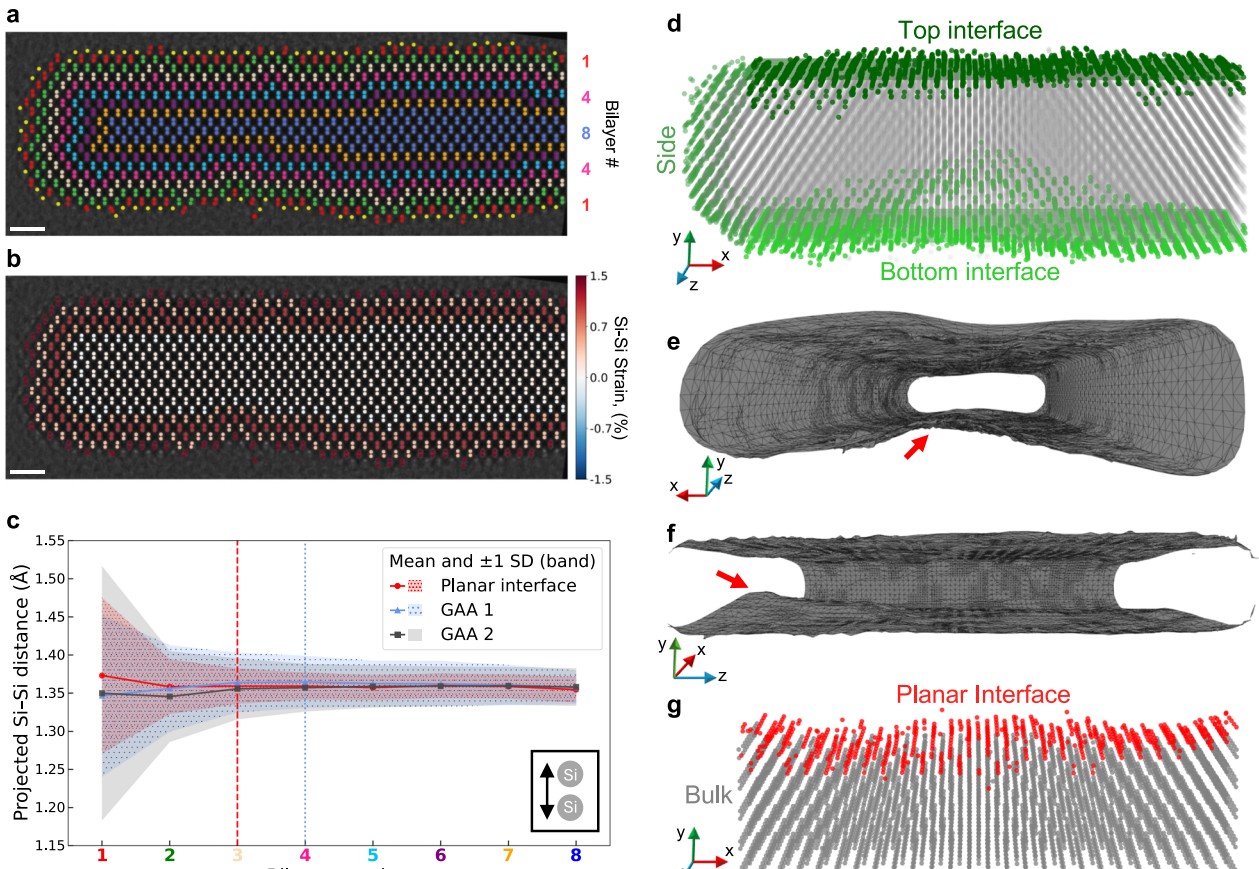

**Fig. 6 | Strain transition along with interface-to-bulk transition and 3D morphology of a GAA structure. a** Tracked atoms overlaid on a depth slice ($z = 14$ nm), with layer-numbering convention for Si bilayers. Colors indicate each pair's relative distance from the c-Si/a-SiO$_2$ interface, where red denotes the interfacial layer. Single detected atoms are shown in yellow. Scale bar: 10 Å. **b** Recolored slice from **a**, with layers colored based on the average Si–Si distance, converted to strain and shown in the color bar. The first four Si bilayers (eight Si atoms) adjacent to the interface are visible as a distinct region in this representation. Scale bar: 10 Å. **c** Projected Si–Si distances as a function of layer number from a planar interface (red circles) and two GAA structures (blue triangles: GAA-1, gray squares: GAA-2). Solid symbols denote the mean Si–Si distance at each layer, while the shaded regions indicate the standard deviation. The x-axis is colored per the layer scheme in **a**. Vertical dashed lines mark the layer

number at which Si–Si distances approach bulk values for the planar (red-dashed vertical line, 3 bilayers) and GAA (blue-dashed vertical line, 4 bilayers) cases. The planar interface reaches bulk-like Si–Si distances by the third bilayer, whereas the GAA structures reach similar values by the fourth bilayer and exhibit larger spread in the measured distances across layers. **d** 3D rendering of GAA-2, showing the full channel morphology. Top, side, and bottom interfaces are color-coded in different shades of green, showing interface variations and a pronounced bottom-interface intrusion. **e, f** Orthogonal views of the GAA-2 surface model, highlighting the same bottom intrusion and showing the extent of interface roughness along the channel. **g** 3D rendering of the planar interface for comparison, with the interfacial tracked atoms in red, illustrating its flatter morphology and reduced roughness relative to the GAA sections (for quantitative comparison, see Supplementary Fig. 8).

studied here, the shortest direction in the Si channel is around 50 Å, with top and bottom interfaces, this means that over 40% of the silicon in the channel is under strain (Fig. 6b, c)—an important consideration for mobility and performance.

Conventional (S)TEM approaches have long struggled to resolve buried interface roughness in 3D, leading to conflicting models: some suggesting exponential height distributions[24], while others favor Gaussian fits[22,64] even though neither form had strong experimental evidence or a clear physical basis[15,21]. In both roughness models two fitting parameters are used: RMS amplitude $\Delta$ and correlation length $\Lambda$, extracted by fitting the autocorrelation function of the measured roughness with exponential or Gaussian models (see Methods and Supplementary Fig. 8). Exponential profiles correspond to abrupt, low-correlation roughness, while Gaussian profiles indicate smoother, long-range variations, reflective of the interface's growth history. However, such detail about the actual interface profile has not previously been accessible with conventional (S)TEM techniques, as can be seen in Fig. 2e–g and Supplementary Fig. 4, where both tf-iDPC and tf-ADF fail to correctly capture the nuanced interface shape. With MEP,

we can now directly resolve buried interfaces in 3D (Fig. 6d–e, Supplementary Movies 4–5) and determine not only the amplitude $\Delta$ and correlation length $\Lambda$ but also the functional form of the roughness distribution at atomic-scale resolution in nanoscale devices rather than macroscopic planar films. Importantly, we find that different interfaces can follow different distributions, highlighting that no single functional form universally applies and underscoring the need for direct atomic-scale measurements of roughness. Supplementary Table 2 presents the measured interface roughness parameters for the reference planar interface and the two GAA interfaces, generated by tracking the outermost atoms of each interface. We find that smoother, intact interfaces primarily with step-edges are best described by exponentially decaying roughness (Supplementary Fig. 8a–f). In contrast, rougher interfaces, such as the one with a mouse-bite, deviate from exponential model and look more Gaussian, indicating the limits of a simple two-parameter model (Supplementary Fig. 8d, g–h). The top and bottom interfaces likely differ due to their different epitaxial histories: the SiGe-on-Si bottom interfaces tend to have mouse-bites and pinholes, while the top Si-on-SiGe interface is

smoother, consistent with strain-driven roughening and Si interdiffusion[65–67] (Fig. 6d–g). Mouse-bites are present in some of the top interfaces too, but less frequently.

These quantitative values of $\Delta$ and $\Lambda$ can be linked to carrier mobility using established scattering formalisms. In the long-wavelength limit, where the carrier wavelength is much larger than the characteristic interface roughness scale $\Lambda$, the carriers experience the interface as a smooth averaged potential, and interface roughness scattering rates scale approximately as

$$\tau^{-1} \propto \Delta^2 \Lambda^2 \qquad (3)$$

and since mobility ($\mu$) is proportional to the scattering time ($\tau$), mobility scales as

$$\mu \propto \frac{1}{\Delta^2 \Lambda^2} \qquad (4)$$

leading to reduced mobility for rougher interfaces[68]. At shorter wavelengths, the dependence is more complex and requires full evaluation of the roughness power spectrum. Recent master-equation simulations have shown that surface-roughness scattering is in fact the predominant mobility-limiting mechanism in ultrathin-body FETs[21]. Using our measured roughness values and adopting the long-wavelength limit of mobility and interface roughness scaling, we estimate that mobility in these early-stage and defective GAA channels is reduced by approximately 7.5 times for the top GAA interface and 37 times for the bottom GAA interface, compared to the planar reference (Supplementary Table 2). Crucially, our ability to measure interface roughness distribution ($\Delta$ and $\Lambda$) directly at the buried interface now provides a pathway to testing and refining these scattering models in future studies.

Spatially-resolved roughness analysis also complements our simultaneous strain mapping within the same volumetric dataset, enabling the decoupling of strain and morphology. The variations we found show geometry and process-dependent differences in c-Si/a-SiO$_2$ interface quality, which we can now directly measure using MEP. This combined metrology is especially crucial at extreme scaling, where roughness and strain strongly influence channel carrier mobility, threshold voltage, and device reliability. Indeed, recent studies on sub-5 nm silicon nanoribbons attribute performance degradation to precisely these effects[4], which we can now measure directly from a single dataset, opening a path towards more studies to understand these parameters better for more predictive and realistic modeling. Interface roughness, and the need to measure it directly in buried interfaces in 3D is also relevant for quantum devices, where interface disorder leads to charge noise and decoherence, as recently demonstrated for Si/SiGe spin qubits[64].

## Discussion

For the information gained, MEP is relatively time- and resource-efficient: from sample preparation to a fully reconstructed 3D image takes only about 2 days, comparable to standard TEM semiconductor characterization, providing rapid atomic-scale structural feedback. (See Typical workflow duration section in Methods). MEP uses the same sample preparation and geometry as standard STEM imaging and is performed on the same instrument, with only the 4D-STEM detector as an additional hardware requirement. MEP can be applied after any fabrication step during process development, providing early three-dimensional atomic-scale structural feedback well before completion of the full fabrication flow. This information complements short-loop electrical testing, which primarily probes aggregate electrical parameters such as current, capacitance, and resistance, by instead directly characterizing buried structural variations and process-dependent interfacial disorder that are not accessible through electrical measurements alone or through conventional projection-based TEM imaging. By providing this complementary structural information, MEP enables faster learning and refinement of fabrication processes. In fact, with experience and recently published packages, like PtyRAD[60], a preliminary 3D reconstruction of a ~20×20×30 nm volume was done in under an hour. The examined GAA structures were in the early stages of development, with synthesis still non-optimized; as such, the presence of severe defects and interface roughness was expected. Fully-processed, commercially available production devices have also been examined by the Cornell group and do not exhibit these defects or this level of roughness, indicating that optimized processing can eliminate such issues and that structural defects present in prototype structures are absent in fully optimized commercial devices. This study demonstrates that MEP provides atomic-scale 3D access to device performance limiting features, namely interface roughness and strain. Through simulation comparisons and experimental validation, we showed that MEP surpasses conventional STEM imaging techniques in recovering atomic-scale features with sub-Ångström lateral and few-nanometer depth resolution. By directly tracking atomic positions across depths, we measured depth-resolved interface roughness and imaged stacking faults in crystalline silicon channels in prototype GAA structures. We found that the smoother, top interface (formerly SiGe-on-Si) primarily has step-edge-driven roughness which follows an exponential spatial correlation, while the rougher bottom interface (formerly Si-on-SiGe) has more pinholes and exhibits irregular roughness fluctuations—likely tied to its epitaxial growth history. Importantly, we resolved not only the amplitude and correlation length but also the functional form of the roughness distribution at atomic resolution, revealing that different interfaces can follow either exponential or Gaussian distributions, reflecting their different synthesis conditions. In these same devices, we also found that silicon reaches its bulk-like structure only after 11 Å from the c-Si/a-SiO$_2$ interface – over 40% of the channel height – compared to 8 Å in a control planar interface, quantifying process-dependent interface quality.

By providing $\Delta$, $\Lambda$, and the actual distribution shape, MEP now provides the experimental parameters needed to test and refine models of carrier mobility and interface roughness scattering in nanoscale devices. These roughness measurements, combined with simultaneously-measured, spatially-resolved strain mapping, provide a framework to assess interface quality, predict device performance, and guide early defect detection in both classical and quantum devices. As device dimensions approach the atomic scale, these capabilities help close critical metrology gaps, inform predictive modeling, and link process-dependent structural variations directly to transport limitations.

## Methods

### Device fabrication

The GAA devices were fabricated following a customized version of the full electrical device flow developed by imec, with simplifications to enable direct access to critical structural features for characterization. Specifically, the process follows the basic CFET flow[69], modified to produce 3× nanosheet GAA structures using a different starting epitaxial stack. The middle dielectric isolation and Epi source/drain modules were skipped to allow early access to the nanosheet release and gate stack deposition steps. The devices were formed by first growing a superlattice of alternating Si/SiGe single-crystal epitaxial layers. The SiGe layers serve as sacrificial materials that are selectively etched to release the Si nanosheets, which ultimately act as the transistor channels. The resulting gaps are then used to deposit the gate stack (typically a-SiO$_2$/a-HfO$_2$/metal gate). An interfacial a-SiO$_2$ layer is first formed by oxidizing the Si nanosheets, followed by high-k and metal gate deposition via atomic layer deposition (ALD), which offers the precise thickness control and conformality needed to uniformly cover the nanosheets on all sides. This stacked geometry enables each ALD step to coat multiple Si channels simultaneously[3,69–71].

The planar interface (c-Si/a-SiO$_2$/a-HfO$_2$) blanket film sample used for comparison was fabricated separately. The Si portion of this sample corresponds to the starting Si wafer, while the SiO$_2$ layer is the native oxide present on the wafer surface, which serves as an effective nucleation layer for subsequent ALD deposition. The HfO$_2$ film was deposited at 300 °C using ALD with alternating HfCl$_4$ and H$_2$O pulses and N$_2$ purge steps between pulses.

## Electron microscopy sample preparation

The c-Si/a-SiO$_2$/a-HfO$_2$ cross-sectional lamella was prepared using a standard cross-section Focused Ion Beam (FIB) lift-out procedure on Thermo Fisher Scientific Helios G4 UX FIB. The lamella is 10–20 nm thick in the region of interest. The GAA sample was prepared by Eurofins Nanolab Technologies with thickness of ~30–40 nm in the region of interest.

## Data acquisition

All the experimental datasets were acquired using a Cs-corrected Thermo Fisher Scientific Spectra 300 X-CFEG at 300 keV with a probe convergence semi-angle of 30 mrad. The MEP datasets were acquired at 10–15 nm nominal overfocus values, using an EMPAD-G2 detector, with scan step sizes in real space of 0.43 Å or 0.81 Å, scan dwell time of 100 μs, and diffraction space sampling of 0.65 mrad per pixel or 0.82 mrad per pixel. The scan fields of view were ~10–20 nm with 256 × 256 pixel scans. A total dose of 0.5–2.5 × 10$^5$ e$^-$ Å$^{-2}$ was used for the 4D-STEM datasets.

The DPC images were acquired on the Thermo Fisher Scientific Panther detector (8 sections) and the ADF images were acquired using the Thermo Fisher Scientific ADF detector, with total through focal series dose of 1 × 10$^5$ e$^-$ Å$^{-2}$ using an automated acquisition method outlined here[72]. Nominal defocus values were corrected by a factor of two based on calibration. For the through-focal datasets, 20 focal planes were acquired with a defocus step-size of 20 Å. High-pass filtering was applied to the tf-iDPC images to suppress low-frequency artifacts, ensuring consistency with the ptychographic reconstructions. All experimental parameters for both MEP and through-focal imaging are summarized in Supplementary Table 3.

## Ptychography reconstruction

Multislice electron ptychography (MEP) was employed to reconstruct the 3D atomic potential of the sample with sub-Ångström lateral and 3–4 nm depth resolution. MEP solves the inverse problem of structure retrieval using 4D-STEM datasets, where a coherent electron beam is raster-scanned across the sample, recording diffraction patterns at each scan position. These diffraction patterns encode both phase and amplitude information of the transmitted electron waves due to overlap in the diffraction spots, and encode depth information due to the parallax effect, allowing for depth-resolved structural reconstruction.

To model the sample, it is conceptually divided into thin slices perpendicular to the beam propagation direction, and phase retrieval is performed using the maximum-likelihood multislice ptychography algorithm implemented in the `fold_slice` package. The version used in this work includes a tilt propagator addition and is available on Zenodo (https://doi.org/10.5281/zenodo.15882443). This algorithm iteratively refines both the probe and sample structure, using a gradient-based approach and multiple probe modes to account for the partial coherence of the electron beam. The optimal reconstruction parameters – convergence semi-angle, defocus, and sample thickness – were determined using a Bayesian optimization model[73], minimizing data error as the objective function.

The reconstruction used 4 probe modes and >1000 iterations, with a slice thickness of 1 nm and depth regularization of 0.4–0.7 (reconstruction summaries in Supplementary Fig. 9). Since each slice in the reconstruction represents a phase object, the amplitude of each slice should be close to one, with deviations primarily due to scattering beyond the outer collection angle of the detector (thus hafnium oxide looks darker), as is shown in Supplementary Fig. 9. High-pass filtering was applied to the final phase reconstructions to eliminate low-frequency artifacts. Supplementary Fig. 9 provides a visual summary of the object phase and amplitude, with intensity histograms, as well as the different probe modes at the entrance surface for all the experimental datasets considered in this work. The fully-converged ptychographic reconstructions were computed using an in-house cluster, requiring approximately one day for completion.

## Typical ptychography workflow duration

The total measurement time for MEP experiments is comparable to standard high-quality TEM workflows. Focused ion beam (FIB) sample preparation for a < 50 nm lamella typically requires half a day for an experienced user, followed by approximately half a day of microscope time (including alignment and tuning). Each 4D-STEM dataset (256 × 256 scan positions) is acquired in about 7 s on the EMPAD-G2 detector. Reconstruction typically takes from an hour to a day, depending on user familiarity, computational resources, degree of convergence needed for imaging vs quantification, and the algorithm used. These times are representative of our setup and will vary between laboratories.

Reconstruction time scales roughly linearly (with a small offset for setup) with the number of probe positions (i.e., area) and with the number of reconstructed depth slices[60]; however, the iterative reconstruction is efficiently accelerated on GPUs and can be distributed across compute nodes. Thus, reconstruction a large area can be done in a similar time, provided more compute resources are available, enabling practical times for larger volumes and multi-device studies.

## Simulation of electron microscopy data

To evaluate the imaging performance of multislice electron ptychography (MEP) and compare it with other imaging modes, we conducted multislice simulations of scanning transmission electron microscopy (STEM) images and 4D-STEM datasets. To validate the accuracy of MEP, we benchmarked its performance in reconstructing a technologically relevant c-Si/a-SiO$_2$/a-HfO$_2$ interface structure and a pMOS structure[12,62]. The simulations were performed using the abTEM multislice simulation package[61]. The simulation approach provides a controlled environment for assessing resolution, sensitivity, and depth reconstruction accuracy while incorporating experimental conditions such as realistic aberrations and electron dose (through detector noise).

The simulations were conducted under conditions mimicking experimental setups, using a 300 keV acceleration voltage and an aberrated electron probe with a convergence semi-angle of 30 mrad. The simulated probe was subjected to spherical ($C_s = 1$ μm) and chromatic ($C_c = 2.46$ mm) aberrations with an energy spread of 0.4 eV. The chromatic focal blur can be estimated as

$$\Delta z_{chrom} \approx \frac{dE}{E_0} C_c \approx 3.3 nm \tag{5}$$

which is of the order of the expected depth resolution, 4.4 nm, and does not affect the imaging modes drastically. On the other hand, spherical aberration shifts the effective entrance surface location by approximately

$$\Delta z_{sph} \approx \sqrt{C_s \lambda} \tag{6}$$

thus shifting the depth features primarily for the tf-ADF and tf-iDPC images. The total electron dose was kept constant at 2.5 × 10$^5$ e$^-$ Å$^{-2}$ across all imaging methods to ensure a fair comparison and Poisson noise was applied to the diffraction patterns to match experimental

conditions. Further simulation details can be found in Supplementary Table 4.

Simulated (4D-)STEM datasets were generated by real space scan step size of 0.14 Å for tf-iDPC and tf-ADF images and 0.44 Å for c-Si/a-SiO$_2$/a-HfO$_2$ MEP and 0.75 Å for GAA MEP. The extent of the pixelated detector was like experimental conditions with maximum collection angle of around 60 mrad and 128×128 diffraction patterns. Using these datasets, various imaging modes were calculated, including MEP, through focal integrated differential phase contrast (iDPC), and through focal annular dark field (tf-ADF) imaging, results of which are presented in Fig. 3 and Supplementary Figs. 1–3. Chromatic aberration was added by averaging simulations at different defocus values according to a gaussian focal spread. Ptychographic reconstructions of the 4D-STEM datasets were performed using the `fold_slice` package; iDPC images were generated from 4D-STEM datasets by computing DPCx and DPCy images, followed by Fourier integration. ADF images were simulated by integrating intensities in annular detectors. For iDPC, only the bright disk was used (30 mrad) while for ADF the inner and outer angles for integration were 60–150 mrad. Unlike ptychographic reconstruction, iDPC and ADF imaging require multiple through-focal datasets to approximate 3D structural information, increasing the total dose and complexity.

The reconstructed images were compared to the ground truth atomic potential, demonstrating that MEP successfully recovered key structural features, including interface roughness, amorphous intrusions, and Si atomic column extend (see Fig. 3 and Supplementary Figs. 1–3). These features are critical for evaluating the performance and reliability of GAA transistors. Compared to iDPC and ADF, MEP provided improved 3D reconstruction accuracy and dose efficiency, mitigating artifacts associated with multiple scattering and channeling.

## Atom tracking and strain relaxation analysis

Atom positions were identified using the Atomap package[63] (see Supplementary Fig. 7 and Supplementary Movies 1,2 and 4). Custom-written Python scripts were used to segment tracked atoms into layers and analyze interfacial structures (see Supplementary Fig. 7 and Supplementary Movies 1,2 and 4). Si-Si projected nearest neighbor distances were calculated for each layer for all the depths for the datasets discussed in this work (see Fig. 6c). Distance mappings were generated to visualize variations in bonding configurations across the layers (for an example see Fig. 6b). Jupyter Notebooks for this analysis are available on Zenodo (https://doi.org/10.5281/zenodo.15882443).

## Three-dimensional visualization

Tomviz was used for volumetric visualization of atomic cuboids shown in Fig. 1e, Fig. 5b and Supplementary Fig. 6a[74]. The c-Si/a-SiO$_2$ interface shape was constructed using the outer-most tracked Si atoms per layer and a 3D surface was constructed using Open3D python package, results of which are shown in Fig. 6e–f and Supplementary Movies 3 and 5[75]. Jupyter Notebook for this analysis is available on Zenodo (https://doi.org/10.5281/zenodo.15882443). Supplementary Movie 6 was rendered in ParaView using synchronized views of the full 3D volume and Z-slices from the GAA-1 reconstructed dataset[76].

## Interface roughness quantification and mobility estimation

To quantify interface morphology, we compute standard roughness metrics: the root-mean-square (RMS) height Δ and in-plane correlation length Λ, directly from the tracked 3D atomic positions. For each dataset, the outermost interfacial Si atoms were segmented into facet-specific groups: planar, top-GAA, and bottom-GAA interfaces. A reference plane was first fitted to the local atomic surface for each facet using a least-squares approach. Height residuals were then computed as the perpendicular distances of each atom from this fitted plane.

The RMS roughness was defined as the average amplitude of deviation of these residual heights across each facet. To assess the lateral spatial coherence of the roughness, we computed the 2D autocorrelation function (ACF) of the roughness map. The in-plane correlation length Λ was extracted by fitting the ACF envelope with both exponential and Gaussian models:

Exponential decay:

$$C(r) \propto e^{-\frac{r}{\Lambda}} \tag{7}$$

Gaussian decay:

$$C(r) \propto e^{-\frac{r^2}{\Lambda^2}} \tag{8}$$

Both fitting models are presented (Supplementary Fig. 9h, i), as there is longstanding discussion in the literature about which model more accurately represents interfacial roughness. Since our MEP reconstructions yield fully 3D atom-resolved data, we can compute the autocorrelation either radially or along specific in-plane directions (x and z), shown in Supplementary Fig. 8. This enables an analysis of anisotropy in the roughness correlation. Axis-resolved correlation lengths are reported alongside radially averaged values to capture the directional dependence of morphological fluctuations.

These quantitative roughness metrics map directly onto mobility via standard interface-roughness (IR) scattering formalism[14]. In planar channels, a compact and widely used scaling for long-wavelength limit follows the relation given in Eq. (4). A full treatment requires evaluation of the roughness power spectrum and numerical calculation of scattering rates[21]. Thus, smoother (smaller Δ) and less correlated (smaller Λ) interfaces yield higher mobility, all else equal. Using this relation and assuming all other relevant parameters to be the same for all interfaces (i.e., carrier density, effective fields), we estimate relative mobility reductions for the measured planar and GAA interfaces, as detailed in Supplementary Table 2.

To fully capture mobility degradation from interface roughness, one must evaluate the complete roughness power spectrum $S(q)$ (the Fourier transform of the measured autocorrelation) together with the electronic form factor and screening effects that govern how carriers sample the interface. In modern master-equation and simulations, which treat these ingredients self-consistently, interface roughness scattering has been identified as the dominant mobility-limiting mechanism in ultrathin-body FETs[21]. By directly measuring both the RMS roughness amplitude (Δ) and correlation length (Λ) from our 3D atomic reconstructions, we now provide realistic inputs for such models, replacing the common practice of treating roughness as an adjustable fitting parameter. This opens the door to quantitative, predictive tests of process-structure-performance relationships in future device simulations.

## Data availability

The experimental 4D-STEM datasets, through-focal ADF/iDPC image stacks, and reconstructed multislice electron ptychography (MEP) phase volumes used to generate the figures in this study are publicly available on Zenodo at https://doi.org/10.5281/zenodo.15882443[77]. Simulation inputs and results are not included in this repository as they are based on third-party structural models[62] that are not owned by the authors.

## Code availability

The MATLAB reconstruction scripts (fold_slice with tilt-propagator extension), parameter files documenting all acquisition and reconstruction settings, and Jupyter notebooks used for atom tracking, strain mapping, and surface morphology analysis are publicly available on Zenodo at https://doi.org/10.5281/zenodo.15882443[77].

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

## Acknowledgements

S.K., D.A.M. acknowledge funding from TSMC through a Joint Development Project (JDP184087). S.E.Z. acknowledges funding from the Platform for the Accelerated Realization, Analysis, and Discovery of Interface Materials (PARADIM), which is supported by the National Science Foundation under Cooperative Agreement No. DMR–2039380. This work made use of the electron microscopy facility of PARADIM and Cornell Center for Materials Research shared instrumentation facility with Helios FIB supported by NSF (DMR-1539918). The authors also thank Malcolm Thomas, Mariena Silvestry Ramos, Philip Carubia, and John Grazul for technical support and maintenance of the electron microscopy facilities. The authors gratefully acknowledge Michael Givens (ASM), Naoto Horiguchi (imec), Hans Mertens (imec), and Hiroaki Arimura (imec) for providing the Gate-All-Around (GAA) sample used in this study. We thank Jiangtao Zhu and Eurofins Nanolab Technologies for preparing the GAA TEM lamella used in this study. We thank Frieder Baumann for the c-Si/a-$SiO_2$ structural model, Richard Aveyard and Bernd Rieger for the pMOS structural model. S.K. gratefully acknowledges Harikrishnan K.P., Ariana Ray, and Salva Rezaie for training and tutorials on MEP, Dasol Yoon for insightful discussions on multislice simulations, Xiyue Zheng for sharing the automated DPC acquisition code, and Yi Jiang for helpful discussions about MEP. S.K. thanks Lopa Bhatt for developing and sharing a tilt propagator extension to the `fold-slice` code used in this study.

## Author contributions

The research plan was formulated by S.K., T.K.C., V.D.H.H. and D.A.M. G.W. sourced the studied samples. S.K. performed STEM and MEP characterization and analysis (experimental and simulation) under the supervision of S.E.Z. and D.A.M. S.K. wrote the original draft with feedback from all authors. All authors discussed the results and commented on the paper.

## Competing interests

Cornell University (D.A.M.) has licensed the EMPAD hardware to Thermo Fisher Scientific. The other authors declare no competing interests.
