## [Transparent Peer Review file · Nature Communications]

3D Atomic-Scale Metrology of Strain Relaxation and Roughness in Gate-All-Around Transistors via Electron Ptychography

Corresponding Author: Professor David Muller

Version 0:

Reviewer comments:

Reviewer #1

(Remarks to the Author)

This study of a GAA device using multislice electron ptychography is precisely the type of experiment needed to further demonstrate and reinforce why MEP is so powerful, and why it will likely largely replace conventional imaging in the near future. This is clearly laid out by the authors, who compare traditional methods (ADF, iDPC) that do not come close to yielding the same insights from a single dataset. Further, all claims are well validated and supported. Given the timeliness, the manuscript should be accepted and published posthaste.

There a couple of minor points:

“By retrieving the object’s phase.” Perhaps better to refer to this as retrieving the phase shifts introduced by the object’s atomically resolved electrostatic potential? In other words, does “object phase” have a broadly understood meaning?

Ultra-scaled has a dash in some places, and no dash in others.

Reviewer #2

(Remarks to the Author)

In high resolution imaging, the depth of field limitations put a limit on the sample thicknesses that can be imaged. Conventional STEM depth-resolved approaches can produce images with a lateral resolution of 0.66-0.89 Å and a depth resolution of ~95 Å. By changing the focus, the resulting 2D images can be stacked to recover the 3D sample structure. By employing multi-slice electron ptychography (MEP) the authors were able to demonstrate that the lateral resolution improves down to 0.49 Å and the depth of field can be extended down to 64 Å. In addition, MEP achieved this imaging performance with at least 2-times lower dose. In summary, MEP enables imaging of thicker sample structures at a higher resolution and with less dose.

While MEP has been developed by prior works, it is the first application (to the best of my knowledge) for metrology of modern semiconductor devices such as the GAA transistor. The 3D structural information looks impressive and highly useful for the semiconductor industry. The surface roughness of the measured transistor features was used to estimate the effect on the charge carrier mobility, enabling optimization of semiconductor device electronic properties. Overall, the presented results demonstrate the capabilities of electron microscopy for quantitative 3D metrology of interface roughness, strain relaxation and atomic-scale defects.

Given that the semiconductor industry is moving towards 3D transistor architectures such as GAA, the need for such 3D metrology tools will only increase. These measurement techniques are also timely since these new transistors will start being introduced to consumer electronic in the next years. Therefore, I believe such metrology capabilities are relevant to the broader semiconductor industry.

Based on my understanding, the MEP methodology is sound and the improvements compared to state-of-the-art are consistent. I do find that there quite a lot of details missing. The details that are within the manuscript they are hard to extract. For example, one of the primary claims is that MEP provides superior spatial and axial resolution, yet finding quantitative numbers is difficult. The axial resolutions comparison to state-of-the-art methods can only be found within one of the image panels and are not searchable within the text. I think the manuscript would benefit greatly from improved structure and flow. I found it quite difficult to track what is going on and had to jump around the text to find the basic quantitative numbers. For example, in MEP the sample is split into slices or during through-focal imaging, a certain number of slices are acquired. Yet there is no number of how many slices were reconstructed/acquired for each method, making it difficult to compare.

As an imaging method developer I found a lot of details missing, making it difficult to draw conclusions about the imaging performance compared to other methods. But from the point of view of the transistor metrology I found the presented results impressive and convincing. I think the presented results meet the required of Nature Communications given their impact.

Comments

1. Line 42 - at this point I think it would be good to say that most metrologies provide 2D images whereas 3D metrology tools are necessary for the metrology of 3D structures. This way it will be clearer for the reader moving forward.
2. Line 71 - "address the metrology gap highlighted above". Could the authors please explicitly state the gap i.e., the need for both angstrom resolution and 3D imaging capabilities. Otherwise it can be not entirely clear for the reader.
3. Line 72 - "By retrieving the object's phase from overlapping diffraction patterns, MEP provides depth information through post-processing of scanning diffraction data". This description is misleading, since the diffraction patterns themselves don't overlap. Instead the overlap is of the neighboring sample illumination patches, which produce these diffraction patterns. Please rephrase by e.g., the sample is scanned across an electron beam and diffraction patterns are collection from overlapping illumination patches.
4. Line 87 - "Formation of the critical GAA structure is one of the earliest steps in a CMOS process that can take 3-4 months". Are the authors claiming that the lithography of GAA transistors takes 3 months during a single wafer production? Please clarify.
5. Line 88 - "Early structural feedback from MEP on witness wafers". What is a witness wafer? The MEP imaging process requires extraction of a thin lamella, which would also be highly disruptive and destructive to the manufacturing process. Could the authors here explain in more detail how MEP could be integrated into the manufacturing process given the constraints of the technique.
6. Line 99 - "tf-ADF signal is ... proportional to the square of the atomic number" and "tf-iDPC ... yields linear or sublinear atomic number contrast". Comparing figure 4b and 4c, the contrast seems to be reversed. If the main differences are linear and squared contrast relationships, then why one is inversely proportional while the other is directly? I would expect both images to have the same direct contrast relationship. Please elaborate why.
7. Line 107 - "correspond to a lateral resolution of 0.40 Å and a depth resolution of 44 Å". Could the authors please explain why the depth resolution should be 44 Å, but in Figure 3, the resulting resolutions are significantly worse? Moreover, MEP achieves 64 Å depth resolution based on Figure 3, which is worse than the quoted imaging resolution of 44 Å. Could the authors please dedicate a paragraph to discuss these theoretical and achieved imaging performance? The whole article states claims such as "better depth resolution" etc., yet there are no quantitative comparison within the text which would state how much better the imaging performance of MEP is compared to other techniques. At the moment the claims are qualitative and scattered throughout the text. The depth resolutions are only shown in Figure 3e and do not appear in the text, which is strange given the emphasis on how much better the depth sectioning is with MEP.
8. Line 111 and 114 - "these issues", "these limitations". Could the authors phrase the issues/limitations more explicitly. e.g., "the primary issue is multiple scattering... and the with MEP we overcoming multiple scattering through multi-slice ptychography, which models the multiple scattering process."
9. Line 115 - first mention of 4D-STEM dataset, please explain here or earlier. Potentially during the description of ptychography and that it acquires 4D-STEM data.
10. Line 117 - "multiple scattering and channeling effects". I am not familiar what are the channeling effects and how multi-slice ptychography can overcome them. Could the authors explicitly mention these artefacts in the earlier section of the text and explain why multiple-scattering happens and what are channeling effects. Potentially around Line 111. Please be clear on the challenges and how they are solved.
11. Line 128 - "Unlike tf-ADF and tf-iDPC (or tomography approaches), which require multiple scans, only a single 4D-STEM dataset is needed for MEP, which can reduce the total electron dose (Figure 2b-d, Extended Data Figure 3)." Please clarify what the authors mean by "scan", since 4D-STEM is also produced by raster-scanning the sample. The authors should clarify that 3D information in MEP is obtained from the same dataset that would be acquired for a 2D sample structure recovery i.e., depth information basically comes "for free". With other approaches the 3D information is does not come "for free" and instead required acquisition of many 2D images e.g., multiple through-focus slices or rotation angles. Without this information some readers might be confused.
12. Line 141-148 - I am missing details on the imaging methods. How many slices are the authors reconstructing with MEP? How many through focus images are the authors acquiring? This would give directly the expected axial resolution for a given sample thickness. What are the performance differences between MEP compared to the through-focus methods. Could the authors expand?
13. Line 166 - "this enhanced resolution is limited by the small diffraction collection angle required for the ~38 nm thick section, whereas the poorer resolution of ADF and iDPC is primarily dose-limited, due to radiation damage concerns". This explanation is not clear. The collection angle for MEP is higher, which allows acquisition of a wider scattering leading to higher resolution, as one would expect. Plus the authors have dose limitations for ADF and DPC leading to even lesser resolution. My questions would be: is the resolution difference explained by the detection acceptance angles beta and alpha

alone? Or it also requires the additional discussion on radiation dose? Could the authors please discuss the resolution from both aspects.

14. Moreover, is the dose for a 2D image acquisition in both MEP and DPC/ADF is the same? ADF/DPC requires many through-focus slices which multiplies the dose requirements. Did the authors reduce the dose for each of the through-focus slices in DPC/ADF or kept it the same? It would be good to clearly establish at the beginning what are the fundamental data acquisition differences between DPC/ADF vs MEP, in terms of dose, data acquisition etc.

15. Line 196 - The following paragraph discusses strain, but it is not explained how they authors obtain the strain information. The resulting MEP reconstruction is simply the 3D structure of the sample. How is the strain recovered from the acquired data?

16. Line 226 - "In both cases two fitting parameters are used...". Could the authors please elaborate what is actually being fitted? I would assume that the authors are fitting the extracted atoms and their positions using Atomap, but it is just a guess on my part.

17. Line 249 - "In the long-wavelength limit...". Could the authors briefly explain what is the wavelength referring to here and how it relates to the carrier mobility mentioned previously.

18. Line 271 - "Importantly, for the information gained, MEP is relatively time- and resource-efficient: from sample preparation to a fully reconstructed 3D image takes only about two days, providing rapid atomic-scale structural feed". Could the authors expand a bit more on the time distribution for each step? E.g., how much slower is the sample preparation compared to imaging itself and the data reconstruction.

19. Line 271 - The paragraph starting here seems to be related more to the "general discussion" and seems unrelated to the findings within the current section. Consider integrating into the discussion.

Minor comments

1. Line 37 - explain FET abbreviation (Field effect transistor).

2. Line 45 - "X-ray ptychographic tomography" rather than just "X-ray ptychography".

3. Line 54 - "projection" to "violation of the projection approximation".

4. Line 79 - if crystalline-Si is c-Si, please add the abbreviation in brackets, which appears later in the text.

5. Line 81 - could the authors please refer here (and in this paragraph in general) to Figure 1 b such that it is clear what the text is referring to. Otherwise for people not familiar with the GAA structure it is unclear.

6. Line 111 - replace "distortions in contrast" to "contrast distortions".

7. Line 132 - "This enables resolution beyond the diffraction limit imposed by the probe convergence semi-angle α ". The probe convergence does not impose a "diffraction limit", only the wave propagation physics imposes a "diffraction limit".

8. Line 144, 223 - "S/TEM" and "(S)TEM". Please use a consistent naming convention.

9. Line 149 - would be good to mention that "c" stands for crystalline and "a" for amorphous.

10. Line 197 - "depth-resolving power" to "depth-resolving capabilities".

11. Line 223 - "for the first time determining" to "determine"

12. Line 533 - "due to linear atomic number contrast, all sections of the structure are visible." what does this mean exactly?

13. Line 623 - "3D rendering of GAA 2's surface", what is the "2's surface"?

14. Figure 1 b - it is unclear what the text is pointing to exactly. Could the authors highlight and assign the layers better by e.g., a contour for each layer. Otherwise it is unclear.

15. Figure 2 c - "Visualization of depth-resolved slices obtained by different methods". I only see three slices, which I assume are from MEP. Where are the other slices "obtained by different methods" which here should include also through focal series? Description seems misleading.

Reviewer #3

(Remarks to the Author)

Karapetyan et al describe a carefully written study using multislice electron ptychography (MEP) to study the nanoscale local structure of an advanced gate-all-around (GAA) nanosheet transistor. GAA transistor structures are central to next generation microelectronics. Characterizing the nanoscale structure of these nanodevices is a critical metrology challenge for the microelectronics industry. This report describes how MEP can provide nanoscale interface roughness and directly measure the local strain in the silicon channel of a single GAA transistor. They compare MEP to other electron microscopy modes and show how it improves what can be extracted from the microscope. Below I detail a couple of minor improvements that can be made to the manuscript. The manuscript will be of wide interest to the readership of Nature Communications.

Seeing this level of detail on a single device is amazing and will be of great interest to the semiconductor industry. The authors should note that a chip has billions of transistors and that there is often variation between each individual device. That variation is quite important for product performance. The unaddressed point is statistical sampling. Measuring one out of a population of one billion is not statistically representative of the population. If they measured a second transistor it likely would be different from the first one. This makes the discussion in the manuscript about speeding up the measurement and reconstruction algorithm critical to the utility of the measurement so that it can be applied to many transistors to improve the statistics. I should note that for the purposes of the manuscript it is acceptable that they only report on one transistor device.

It would be helpful to non-microscopists to include some additional diagrams in the figures to help one understand where in the device the particular view/slice comes from. The slices sometimes can be non-intuitive, especially with the asymmetric resolution. For example, on figure 1 it would be helpful to add marks on the 3D image in 1C highlighting where in z the slices shown in 1d come from. They should be color coded better. Making 1D the same green as 1C is a little confusing.

Extended data figure 4 - not clear from the caption what is different between the two images. The figures also appear to have

different scalings in the two directions. The vertical dimension is 20 nm according to the arrows and the horizontal scale bar is 1 nm. The difference in scaling is about 3.67x.

The methods section does not describe the fabrication of the silicon, silicon oxide, and hafnium oxide blanket film.

I recommend mentioning in the abstract that the GAA devices are fabricated by imec. With the TSMC coauthors and no imec coauthors it is easy to assume the devices were made by TSMC.

Version 1:

Reviewer comments:

Reviewer #2

(Remarks to the Author)

The authors have addressed all of my comments and have made substantial improvements to the manuscript. I greatly appreciate the hard work of the authors and would be happy to see this work published. I have nothing to add or critique at this stage.

Reviewer #3

(Remarks to the Author)

The authors have provided an extremely thorough and careful response to the reviews. I thank them for that. The clarity of the manuscript has improved a lot. I think it now would be clear to both TEM people not familiar with advanced semiconductor devices and semiconductor people that are not TEM experts. There are a few minor issues that would further improve the manuscript.

In the introduction, the authors distinguish between “light” elements and “heavy” elements and changes in the contrast for different imaging methods associated with atomic number. Later in the manuscript in the caption of figure 1 they state that light for GAA devices means Si and O and heavy is Hf. It would be valuable to say that in the main text where light and heavy are discussed in the introduction (page 4).

Paragraph starting on line 74, page 4. Somewhat redundant multiple uses of “light and heavy elements” and sub-Angstrom resolution (lines 75-77 and 81-83). “the need for Angstrom-scale resolution three-dimensional imaging capabilities, and simultaneous sensitivity to both light and heavy atoms” and “MEP has been established as a technique capable of sub-Angstrom in-plane resolution and nanometer-scale depth resolution, with sensitivity to both light and heavy elements.” They can combine those sentences to smooth the flow.

Page 9 (lines 195-199) – The transition between simulated and real data is not clear in the text. They do clearly state that the GAA is experimental, but it would be good to say when they change from silicon/silicon oxide/hafnium oxide to the Ta-filled pinhole sample that it is still simulated microscopy.

Figure 4c – According to the caption, the 3rd slice in 4c corresponds to the single atom marked by an arrow in 4b. The other slices correspond to single atoms shown in the supplemental. I recommend they put a red box around the slice in 4c that corresponds to the red arrow in 4b.

On page 5 (lines 95-100) and page 15 (lines 328-329) the manuscript mentions electrical testing would require months of fabrication to produce a testable device. Full production chip take months to produce, but during process development fabs make short loop wafers with a small number of metal layers to allow function testing after a much shorter time (days to weeks). This is critical for development of new device structures/processes since at that time they have no idea how to relate images of the devices to end performance. The current wording is a little misleading since no fab would take multiple months to be able to etest structures during new device development. The wording seems to merge process monitoring during production where an early metrology step could save a lot of time/cost and metrology during process development where it helps them know what they made and relate structure to electrical function. Separate from etest, 3D atomic scale resolution would speed up the development cycle by speeding up learning.

How does computation time scale with the size of the measured volume? Since the 4D STEM measurement is pretty fast, end users would be interested in trying to get a larger reconstruction volume.

Typo (page 8, line 176) – “rough crystalline-silicon (c-Si)/amorphous hafnium oxide (a-SiO₂)/amorphous-hafnium oxide (a-HfO₂) interface.” The first hafnium oxide should be silicon oxide.

The author’s general response to reviewer #3 about measurement statistics appears to mix up two different uses for 3D atomic scale resolution imaging – failure/fault analysis and development testing/feedback. Both are important. It is very difficult to make what you can’t measure, so fast data on process changes speed up the learning loop. In the case of development testing/feedback statistics is very important because they aren’t trying to look at a specific transistor to figure out why it didn’t work. They are trying to learn what changed in the transistors for a processing variation. None of the review response discussion about failure analysis was in the manuscript so the comment is only to correct the review response.

Version 2:

Reviewer comments:

Reviewer #3

(Remarks to the Author)

I thank the authors for their updates. The manuscript is now ready for publication. Their method will add a very powerful tool to the toolbox of the semiconductor industry.

Responses to Reviewer Comments:

We thank all three reviewers for their thoughtful and constructive feedback. We appreciate the reviewers' positive assessment of the significance of MEP within the semiconductor metrology toolbox, and we have revised the manuscript accordingly to improve clarity, completeness, and reader accessibility. We have carefully addressed each comment in the point-by-point responses below. Our responses are shown in **green** and revised or newly added text in the manuscript is shown in **blue**, with the relevant modifications in **bold**. All line numbers referenced in our responses correspond to the revised clean draft (with tracked changes accepted), which is provided separately from the marked-up version.

Below is a summary of the key changes:

To address the reviewers' suggestions regarding clarity and structure, we have re-organized several sections of the manuscript to better match the typical format of a Nature Communications article. This includes restructuring the imaging-results and analysis sections for improved logical flow and adding the previously missing Code Availability statement. We have also expanded the Methods section and added new tables in the Supplementary Information that summarize all imaging and reconstruction parameters.

Several figures have been revised for clarity and completeness. In particular, Figure 3 has been updated to clearly distinguish the apparent depth extent from the depth blur, reducing the potential for misinterpretation. We have moved certain figures earlier in the manuscript where they more naturally support the surrounding discussion, added quantitative analyses of axial blur for the different imaging modes, and consolidated all depth-resolution values in Supplementary Table 3. We have also included an explicit experimental estimate of the MEP depth blur. We hope these updates ensure that the quantitative performance of MEP relative to tf-ADF and tf-iDPC can be located quickly and unambiguously.

REVIEWER COMMENTS

Reviewer #1 (Remarks to the Author):

This study of a GAA device using multislice electron ptychography is precisely the type of experiment needed to further demonstrate and reinforce why MEP is so powerful, and why it will likely largely replace conventional imaging in the near future. This is clearly laid out by the authors, who compare traditional methods (ADF, iDPC) that do not come close to yielding the same insights from a single dataset. Further, all claims are well validated and supported. Given the timeliness, the manuscript should be accepted and published posthaste.

We sincerely thank the reviewer for their very positive and encouraging assessment of our work, and for recognizing both the significance and timeliness of applying Multislice Electron Ptychography (MEP) to advanced GAA transistor metrology. We are grateful for the reviewer's supportive comments affirming the validity of our results and the potential of MEP to transform atomic-scale imaging in semiconductor research.

Below, we provide point-by-point responses addressing the reviewer's comments.

There is a couple of minor points:

Comment #1:

"By retrieving the object's phase." [OBJ] Perhaps better to refer to this as retrieving the phase shifts introduced by the object's atomically resolved electrostatic potential? In other words, does "object phase" have a broadly understood meaning?

Response:

We agree and have revised the text for clarity (lines 79-81):

"... MEP retrieves the **phase shift introduced by the object's atomically-resolved electrostatic potential**, providing depth information through post-processing of scanning diffraction data."

Comment #2:

Ultra-scaled has a dash in some places, and no dash in others.

Response:

Thank you for catching this, we have changed all to be "ultrascaled" with no dash.

Reviewer #2 (Remarks to the Author):

In high resolution imaging, the depth of field limitations put a limit on the sample thicknesses that can be imaged. Conventional STEM depth-resolved approaches can produce images with a lateral resolution of 0.66-0.89 Å and a depth resolution of ~95 Å. By changing the focus, the resulting 2D images can be stacked to recover the 3D sample structure. By employing multi-slice electron ptychography (MEP) the authors were able to demonstrate that the lateral resolution improves down to 0.49 Å and the depth of field can be extended down to 64 Å. In addition, MEP achieved this imaging performance with at least 2-times lower dose. In summary, MEP enables imaging of thicker sample structures at a higher resolution and with less dose.

While MEP has been developed by prior works, it is the first application (to the best of my knowledge) for metrology of modern semiconductor devices such as the GAA transistor. The 3D structural information looks impressive and highly useful for the semiconductor industry. The surface roughness of the measured transistor features was used to estimate the effect on the charge carrier mobility, enabling optimization of semiconductor device electronic properties. Overall, the presented results demonstrate the capabilities of electron microscopy for quantitative 3D metrology of interface roughness, strain relaxation and atomic-scale defects.

Given that the semiconductor industry is moving towards 3D transistor architectures such as GAA, the need for such 3D metrology tools will only increase. These measurement techniques are also timely since these new transistors will start being introduced to consumer electronic in the next years. Therefore, I believe such metrology capabilities are relevant to the broader semiconductor industry.

Based on my understanding, the MEP methodology is sound and the improvements compared to state-of-the-art are consistent. I do find that there quite a lot of details missing. The details that are within the manuscript they are hard to extract. For example, one of the primary claims is that MEP provides superior spatial and axial resolution, yet finding quantitative numbers is difficult. The axial resolutions comparison to state-of-the-art methods can only be found within one of the image panels and are not searchable within the text. I think the manuscript would benefit greatly from improved structure and flow. I found it quite difficult to track what is going on and had to jump around the text to find the basic quantitative numbers. For example, in MEP the sample is split into slices or during through-focal imaging, a certain number of slices are acquired. Yet there is no number of how many slices were reconstructed/acquired for each method, making it difficult to compare.

As an imaging method developer I found a lot of details missing, making it difficult to draw conclusions about the imaging performance compared to other methods. But from the point of view of the transistor metrology I found the presented results impressive and convincing. I think the presented results meet the required of Nature Communications given their impact.

We sincerely thank the reviewer for their careful reading of the manuscript, their positive assessment of our results, and their recognition of the broader relevance of multislice electron ptychography (MEP) for semiconductor metrology. We greatly appreciate the reviewer's detailed engagement with both the technical aspects of the imaging methodology and the clarity of the manuscript. Their comments have helped us significantly improve the presentation, accessibility, and completeness of the work.

In response to the reviewer's request for clearer structure and for quantitative imaging performance metrics to be more readily identifiable, we have substantially revised the manuscript for improved flow and readability. These revisions include:

- Adding measurements of depth resolution, both for simulation and experiment.
- Adding a paragraph clarifying the theoretical depth limits, the achieved simulated and experimental depth resolution, and the distinction between apparent thickness and true axial resolution.
- Introducing Supplementary Table 3, which consolidates all theoretical and experimental depth-resolution values for MEP, tf-ADF, and tf-iDPC in one searchable location.
- Expanding our explanation of probe-sample interactions (multiple scattering, channeling) and how these degrade conventional through-focal methods but are explicitly modeled and corrected in MEP.
- Summarizing acquisition and reconstruction parameters in Supplementary Table 1 (noting that our Zenodo repository had the datasets with the full metadata).
- Improving the flow in the Results sections so that definitions, numerical values, and experimental conditions appear in a logical order without requiring the reader to jump between sections.
- Including new figure annotations, improved captions, and a rewritten benchmarking section for clearer comparison of MEP with tf-ADF and tf-iDPC.

We are grateful for the reviewer's careful reading and constructive feedback, which substantially improved the manuscript. Below we respond point-by-point to all comments.

Comments

1. Line 42 - at this point I think it would be good to say that most metrologies provide 2D images whereas 3D metrology tools are necessary for the metrology of 3D structures. This way it will be clearer for the reader moving forward.

Response:

We thank the reviewer for the helpful suggestion. We have added a brief statement highlighting that most conventional metrologies provide 2D information, whereas novel 3D metrology is required to fully characterize modern 3D atomic-scale semiconductor structures (lines 43-46):

“Various imaging techniques have been employed to characterize semiconductor devices over the decades, **most of which provide only two-dimensional information, whereas 3D metrology tools are necessary to fully characterize the atomic-scale structure and complexity of modern three-dimensional device architectures.**”

2. Line 71 - "address the metrology gap highlighted above". Could the authors please explicitly state the gap i.e., the need for both angstrom resolution and 3D imaging capabilities. Otherwise it can be not entirely clear for the reader.

Response:

We have revised the text to explicitly state the metrology gap identified by the roadmaps as the need for Ångström-scale resolution, three-dimensional imaging capabilities, and the ability to simultaneously resolve both light and heavy atoms, enabling direct measurement of atomic structure, strain, and interface roughness in 3D (lines 74-78):

“Here, we show how multislice electron ptychography (MEP) addresses the critical metrology gap in device characterization highlighted above, **namely, the need for Ångström-scale resolution, three-dimensional imaging capabilities, and simultaneous sensitivity to both light and heavy atoms, thereby enabling direct measurement of atomic structure, strain, and interface roughness in 3D.**”

3. Line 72 - "By retrieving the object's phase from overlapping diffraction patterns, MEP provides depth information through post-processing of scanning diffraction data". This description is misleading, since the diffraction patterns themselves don't overlap. Instead the overlap is of the neighboring sample illumination patches, which produce these diffraction patterns. Please rephrase by e.g., the sample is scanned across an electron beam and diffraction patterns are collection from overlapping illumination patches.

Response:

We have revised the sentence to clarify that the overlap occurs in the illuminated regions of the sample rather than in the diffraction patterns themselves (lines 78-81):

“**By scanning a convergent electron probe across the sample and collecting diffraction patterns from overlapping illuminated regions, MEP retrieves the phase shift introduced by the object's atomically resolved electrostatic potential, providing depth information through post-processing of scanning diffraction data²⁶.**”

4. Line 87 - "Formation of the critical GAA structure is one of the earliest steps in a CMOS process that can take 3-4 months". Are the authors claiming that the lithography of GAA transistors takes 3 months during a single wafer production? Please clarify.

Response:

The 3-4 months refers to the total duration of the complete CMOS fabrication process, which involves about a thousand steps. We have revised the text to clarify that GAA structure formation occurs early in this overall process (lines 95-97):

"Formation of the critical GAA structure occurs early in the overall CMOS fabrication process, which comprises roughly a thousand steps and typically takes 3-4 months to complete from start to finish."

5. Line 88 - "Early structural feedback from MEP on witness wafers". What is a witness wafer? The MEP imaging process requires extraction of a thin lamella, which would also be highly disruptive and destructive to the manufacturing process. Could the authors here explain in more detail how MEP could be integrated into the manufacturing process given the constraints of the technique.

Response:

A witness wafer refers to a dedicated test wafer fabricated in parallel with production wafers for process monitoring and metrology for exactly these concerns. We have clarified this in the text (lines 97-100):

"Early structural feedback from MEP performed on dedicated witness wafers (test wafers fabricated alongside production wafers for process monitoring, including by destructive analysis techniques) can provide rapid insight into structural defects and interface quality early in fabrication flow and reduce costly iterations."

While MEP requires lamella extraction, such destructive analysis is already standard practice for structural characterization in semiconductor process development and is routinely performed on witness wafers without affecting production yield.

6. Line 99 - "tf-ADF signal is ... proportional to the square of the atomic number" and "tf-iDPC ... yields linear or sublinear atomic number contrast". Comparing figure 4b and 4c, the contrast seems to be reversed. If the main differences are linear and squared contrast relationships, then why one is inversely proportional while the other is directly? I would expect both images to have the same direct contrast relationship. Please elaborate why.

Response:

This reflects a breakdown of the standard imaging approximations used to describe these imaging modes. The statements about tf-ADF and tf-iDPC contrast ($\sim Z^2$ vs \sim linear/sublinear) refer to their ideal behavior for very thin (a few nm) samples, where the electron probe shape remains essentially unchanged as it propagates through the

specimen, so the resulting image can be interpreted directly in terms of the projected atomic potential of the sample.

In our experimental data (revised Figure 2e-g), however, the GAA lamella is tens of nanometers thick (~38nm), well beyond the weak-interaction regime, which holds only for samples a few nanometers thick, since electrons interact strongly with matter. At this thickness, the electron probe undergoes multiple scattering and electron channeling (described in a newly added paragraph, also as per comment 10), which modify the wavefront as it traverses the sample. These effects lead to deviations from the ideal contrast relations and can even invert contrast in iDPC. Such behavior is well documented [1-3] and is one of our primary motivations for employing MEP, which explicitly models these interactions, avoiding such artifacts.

To clarify this, we have added two new paragraphs that explicitly note that the linear or quadratic Z-dependence applies only in the thin-sample limit (for iDPC and ADF, respectively) and that deviations arise in realistic, thicker device specimens due to multiple scattering and channeling, which are explained in the following paragraph and cause the artifacts in iDPC contrast that the reviewer is pointing out (starting on line 116):

“.... tf-ADF uses high-angle scattering (Figure 2a, blue), which, **in very thin samples**, exhibits approximately Z^2 -dependent contrast (Figure 2f). tf-iDPC uses low-angle scattering (Figure 2a, purple) to approximate the projected potential, yielding, **in very thin samples**, linear or sublinear atomic number contrast and improved sensitivity to light elements (Figure 2g)³⁶.

Both methods rely on a linear, weak-phase imaging approximation—i.e., that the probe shape remains unchanged from its free-space solution as it propagates through the specimen—an assumption that fails for device-relevant thicknesses (tens of nanometers) due to multiple scattering. At these thicknesses, the probe wavefront is repeatedly altered as it interacts with successive atomic planes (multiple scattering) and can be guided along the positively-charged columns of atomic nuclei (channeling)³⁷, breaking the direct relationship between image intensity and the underlying atomic potential. These nonlinear probe-sample interactions produce well-documented artifacts^{10,11,30,36}, **including axial elongation, contrast reversals, and positional inaccuracies, which lead to mis-localized interfaces and spurious depth-dependent features (Figure 2e–g)."**

1. Bosch, E. G. & Lazić, I. Analysis of depth-sectioning STEM for thick samples and 3D imaging. *Ultramicroscopy* **207**, 112831 (2019).
2. Lazić, I., Bosch, E. G. T. & Lazar, S. Phase contrast STEM for thin samples: Integrated differential phase contrast. *Ultramicroscopy* **160**, 265–280 (2016).

3. Bürger, J., Riedl, T. & Lindner, J. K. N. Influence of lens aberrations, specimen thickness and tilt on differential phase contrast STEM images. *Ultramicroscopy* **219**, 113118 (2020).
7. Line 107 - "correspond to a lateral resolution of 0.40 Å and a depth resolution of 44 Å". Could the authors please explain why the depth resolution should be 44 Å, but in Figure 3, the resulting resolutions are significantly worse? Moreover, MEP achieves 64 Å depth resolution based on Figure 3, which is worse than the quoted imaging resolution of 44 Å. Could the authors please dedicate a paragraph to discuss these theoretical and achieved imaging performance? The whole article states claims such as "better depth resolution" etc., yet there are no quantitative comparison within the text which would state how much better the imaging performance of MEP is compared to other techniques. At the moment the claims are qualitative and scattered throughout the text. The depth resolutions are only shown in Figure 3e and do not appear in the text, which is strange given the emphasis on how much better the depth sectioning is with MEP.

Response:

As discussed below we have now added measurements and discussion of depth resolution. What we had in Figure 3e was not meant to represent the depth resolutions themselves. Instead, they show where a known 66 Å-thick Si column appears when imaged using each method. The question being evaluated is: given a true axial thickness of 66 Å, what thickness and apparent depth location of the column does each method recover? MEP correctly reconstructs 66 Å, whereas tf-ADF and tf-iDPC return ~93–94 Å, reflecting axial elongation and surface-shift artifacts as a consequence of the probe-sample coupling and non-linear propagation discussed above. MEP has properly accounted for these effects. We have changed the text to make this clearer (lines 183-186).

To estimate the depth resolution (or in this case the axial point-spread width), we fitted error functions to the leading and trailing edges of each axial profile in Figure 3e, essentially a measure of the edge-spread function. Below is the updated text (lines 187-194):

“To estimate depth blurring, we fitted error functions to the entry and exit edges of the axial line profiles (see Figure 2e and Supplementary Figure 2). MEP shows an edge blur of only 22 Å, while infinite dose tf-ADF shows 40 Å. The finite-dose tf-ADF profile is too noisy to fit, mirroring the experimental situation. tf-iDPC shows a 57 Å edge blur, comparable to the full sample thickness, indicating that the column boundaries cannot be meaningfully localized. Supplementary Figure 2 shows that MEP’s depth-sectioning performance remains stable across depth-slice spacings and detector collection angles, provided Nyquist sampling is met, and sufficient intensity is captured at the highest collected scattering angles.”

We have remade Figure 3e to include the depth-resolution and made clearer annotations: new gray dashed lines mark the true sample thickness, and the red-shaded regions indicate the edge blurring of each method, so that the distinction between ground truth, apparent thickness, and depth blur is more apparent. These quantitative values are now explicitly included in the main text (lines 188-191) and the revised Figure 3e and its caption (lines 422-431):

“Figure 3. ... e Depth-intensity profiles of the \times -marked c-Si column compared with the true axial extent of the 66 Å-thick Si segment (top). The grey dashed lines show the true positions of the entrance and exit surfaces, while the solid grey lines show the apparent location of these surfaces for the different methods and the corresponding column widths. The red-shaded edge-widths quantify the edge blur (depth broadening) from error-function fits to each method. Because the noisy tf-ADF signal cannot be reliably fitted (blue crosses), the tf-ADF blur values are extracted from the noise-free (infinite-dose) tf-ADF signal (blue line). MEP yields the smallest edge blur (22 Å), while tf-ADF (40 Å) and tf-iDPC (57 Å) exhibit substantially larger axial broadening, consistent with their reduced ability to localize features in depth. Supplementary Figure 1 provides selected depth slices

comparing the three imaging techniques.”

In revising Figure 3 and expanding the discussion of depth resolution, we identified an indexing error in the plotting script that made the true 66 Å Si segment appear as ~58 Å in the earlier figure. The underlying multislice simulations and all reconstructions were unaffected; only the displayed reference profile was mis-indexed. After correcting the visualization and re-running the automated fits, the apparent thickness values shift slightly: MEP changes from 64 Å to 66 Å, tf-iDPC from 95 Å to 93 Å, and tf-ADF from 93 Å to 94 Å. These 1-2 Å differences simply reflect which pixel of the column is selected for the width measurement; we have now standardized this by consistently identifying the brightest pixel in the column. The corrections do not alter any conclusions but now allow us to present a clearer quantitative comparison of depth performance across the three methods.

We have also performed a direct experimental estimation of the MEP depth blur. Unlike the simulated test case, we cannot assume the experimental dataset contains truly “sharp” interfaces: the lamella surfaces are affected by FIB damage, amorphization, and redeposition, making column-edge-based measurements unreliable. Instead, we

measured isolated single atoms (a consequence of redeposition during the FIB milling) present at the entrance and exit surfaces of the MEP reconstructions and extracted their axial intensity profiles. Gaussian + linear-background fits to these peaks, presented in Figure 4 and a new Supplementary Figure 5, yield a consistent depth blur of $\Delta z = 40 \pm 7$ Å (s. d.) across six measurements. We have added text to the main part of the paper that discusses this (lines 207-208).

Equivalent experimental depth-blur measurements could not be performed for tf-ADF or tf-iDPC. Under the experimental dose conditions, the atomic columns in the through-focal series are too noisy to localize reliably, and surface atoms are not stably positioned between defocus frames due to beam-induced motion (lines 208-212). We were not able to increase the dose without damaging the sample, again illustrating the limitations of these methods.

For ease of reference, we have now added **Supplementary Table 3**, which consolidates all theoretical and experimental depth-resolution values in one place:

Supplementary Table 3. Depth-resolution comparison for MEP and through-focal imaging. The theoretical values correspond to multislice simulation-based estimates shown in **Figure 3**. Experimental depth resolutions were measured from the reconstruction shown in Figure 5 and **Supplementary Figure 5**. Through-focal ADF and iDPC did not yield experimental depth-resolution estimates because the through-focal series were too noisy to localize individual atomic columns reliably.

	Theory	Experiment
MEP	22	40 ± 7 Å
tf-ADF	40	-
tf-iDPC	57	-

Although broader than the simulated MEP blur (22 Å for the idealized simulated c-Si test column), this is expected from the stronger regularization setting (0.5) that was used in the experimental reconstructions compared to 0.1 used in the simulation ones, to ensure convergence of the reconstructions, which inherently degrades the depth resolution (lines 222-224).

(See also Responses 12–14 for acquisition and reconstruction parameters and depth resolution constraints imposed by β , Nyquist sampling, dose, and regularization.)

- Line 111 and 114 - "these issues", "these limitations". Could the authors phrase the issues/limitations more explicitly. e.g., "the primary issue is multiple scattering... and

the with MEP we overcoming multiple scattering through multi-slice ptychography, which models the multiple scattering process."

Response:

We have removed the sentence on line 111 that started with "these issues", since now there is discussion of multiple scattering and channeling explicitly (see response to comment 10). We have rephrased the sentence to now read (lines 131-135):

"MEP uses the same optical configuration as tf-ADF and tf-iDPC (Figure 2a) but overcomes their limitations—**probe-sample coupling and the lack of any correction for multiple scattering in thicker specimens**—by recording a momentum-resolved scattering distribution and reconstructing both the probe and the sample potential^{38,39}. The multiple scattering is accounted for using a multislice forward model^{28,40,41} (Figure 2b)."

9. Line 115 - first mention of 4D-STEM dataset, please explain here or earlier. Potentially during the description of ptychography and that it acquires 4D-STEM data.

Response:

We have now clarified what is meant by a 4D-STEM dataset at its first mention and have briefly described how it is acquired (lines 135-138):

".... **In practice, this is achieved by acquiring a four-dimensional scanning transmission electron microscopy (4D-STEM) dataset: a defocused probe is raster-scanned over the sample, ensuring overlap between adjacent illuminated regions, and recording a two-dimensional diffraction pattern at each probe position in real space.**"

10. Line 117 - "multiple scattering and channeling effects". I am not familiar what are the channeling effects and how multi-slice ptychography can overcome them. Could the authors explicitly mention these artefacts in the earlier section of the text and explain why multiple-scattering happens and what are channeling effects. Potentially around Line 111. Please be clear on the challenges and how they are solved.

Response:

We have added a new sentence that explicitly defines multiple scattering and electron channeling and describes the imaging artifacts they produce (see also response to comment 6; text lines 121-130):

"Both methods rely on a linear, weak-phase imaging approximation—i.e., that the probe shape remains unchanged from its free-space solution as it propagates through the specimen—an assumption that fails for device-relevant thicknesses (tens of nanometers) due to multiple scattering. At these thicknesses, **the probe wavefront is repeatedly altered as it interacts with successive atomic planes (multiple scattering) and can be guided along the positively-charged columns of atomic nuclei (channeling)**³⁷, breaking the direct relationship between image intensity and the underlying atomic

potential. These nonlinear probe-sample interactions produce well-documented artifacts^{10,11,30,36}, including axial elongation, contrast reversals, and positional inaccuracies, which lead to mis-localized interfaces and spurious depth-dependent features (Figure 2e–g).”

We have also expanded our explanation on how MEP inherently accounts for these effects (lines 138-142):

“The MEP reconstruction from these datasets explicitly accounts for the probe shape and its evolution—including multiple scattering, channeling, partial coherence, and sample tilts—by iteratively propagating the recovered probe through successive slices of the specimen (Figure 2c), enabling quantitatively accurate 3D reconstructions even for ~40 nm-thick specimens.”

11. Line 128 - "Unlike tf-ADF and tf-iDPC (or tomography approaches), which require multiple scans, only a single 4D-STEM dataset is needed for MEP, which can reduce the total electron dose (Figure 2b-d, Extended Data Figure 3)." Please clarify what the authors mean by "scan", since 4D-STEM is also produced by raster-scanning the sample. The authors should clarify that 3D information in MEP is obtained from the same dataset that would be acquired for a 2D sample structure recovery i.e., depth information basically comes "for free". With other approaches the 3D information is does not come "for free" and instead required acquisition of many 2D images e.g., multiple through-focus slices or rotation angles. Without this information some readers might be confused.

Response:

We thank the reviewer for this helpful suggestion, which we have incorporated it in the new text (lines 147-152):

“Unlike through-focal methods (or tomography approaches), which require **multiple separate x-y raster scans, each representing a separate exposure of the sample to the electron beam, only a single x-y scan and its resulting 4D-STEM dataset is needed for MEP, from which 3D structural information is recovered (Figure 2b-d). In this sense, the depth information in MEP comes essentially “for free” with the x-y raster (Figure 2e), while the use of a single dataset also reduces the total electron dose (Figure 2b-d).**”

12. Line 141-148 - I am missing details on the imaging methods. How many slices are the authors reconstructing with MEP? How many through focus images are the authors acquiring? This would give directly the expected axial resolution for a given sample thickness. What are the performance differences between MEP compared to the through-focus methods. Could the authors expand?

Response:

We now mention that these imaging conditions are described in the methods section on new line 202. We have also provided a new summary table that details all the experimental conditions for ease of comparison (Supplementary Table 1). All the experimental datasets used in this study are also provided in our Zenodo repository together with detailed experimental conditions.

As described in the MEP Reconstruction subsection of the Methods, in this work, the MEP reconstructions were performed using a 1 nm slice thickness. Given the experimentally realized depth blur of approximately 4 nm (see Response 7), this slice spacing satisfies the Nyquist criterion and ensures accurate sampling of the 3D electrostatic potential. For the ~38 nm-thick GAA lamella, this corresponds to 42 reconstructed slices (we include additional slices to ensure the full structure is encompassed). Importantly, the slice thickness in MEP does not define the achievable axial resolution; as long as the sampling satisfies Nyquist, the axial resolution is instead determined by the probe convergence angle, detector collection angle, and signal-to-noise ratio. We also included a brief supplementary comparison showing that using finer vs. coarser MEP slice spacings produces negligible differences for reconstructing the same dataset (Supplementary Figure 2). This simply illustrates that, once Nyquist-sampled, changing the z-slice thickness (or defocus step size) does not improve the achievable axial resolution.

We have also created Supplementary Table 2 that summarizes all the simulation conditions used, again for ease of comparison.

For the through-focal (tf) methods, we have also added to the Methods section the number of defocus planes used in experiments (quoted there for clarity). As with MEP, the defocus step size is chosen to at least Nyquist-sample the expected depth resolution, rather than to set it.

The relative performance of MEP and the through-focal techniques are now discussed in detail in the main text as described in Response 7.

13. Line 166 - "this enhanced resolution is limited by the small diffraction collection angle required for the ~38 nm thick section, whereas the poorer resolution of ADF and iDPC is primarily dose-limited, due to radiation damage concerns". This explanation is not clear. The collection angle for MEP is higher, which allows acquisition of a wider scattering leading to higher resolution, as one would expect. Plus the authors have dose limitations for ADF and DPC leading to even lesser resolution. My questions would be: is the resolution difference explained by the detection acceptance angles beta and alpha alone? Or it also requires the additional discussion on radiation dose? Could the authors please discuss the resolution from both aspects.

Response:

We thank the reviewer for this thoughtful question. We have rewritten the relevant discussion paragraph to more clearly indicate the factors limiting resolution. The achievable resolution in all three imaging modes is set by two main factors: (i) the experimental geometry, which defines the maximum spatial frequencies that can be recorded (set by the probe convergence semi-angle α for tf-ADF/tf-iDPC and by the detector collection angle β for MEP), and (ii) the signal-to-noise ratio, which is limited by the maximum electron dose the sample can tolerate without damage. In practice, the regularization used for the MEP reconstruction (for better convergence) can degrade the achievable depth resolution.

In our experiments, the achievable β for MEP was limited by the $\sim 38\text{--}40$ nm specimen thickness and the finite detector pixel count; increasing β further would require proportionally higher dose to maintain SNR at the largest scattering angles, exceeding the damage threshold, and would undersample the diffraction patterns. These combined geometric and SNR considerations explain the relative resolution performance observed in the experiments.

To further clarify the role of β and sampling, we have added a new supplementary figure comparing simulated MEP reconstructions for $\beta = 3\alpha$ versus 2α and for different slice spacings (5 Å and 10 Å). As shown in Supplementary Figure 2, once Nyquist conditions are satisfied and the available dose is fixed, expanding β beyond 2α yields only marginal improvement under our experimental dose constraints.

Regarding depth resolution, we experimentally estimated the depth blur in MEP by fitting isolated single-atom features, obtaining a consistent FWHM of 40 ± 7 Å. Performing an equivalent analysis for tf-ADF and tf-iDPC was not feasible because individual atoms are too noisy and often mobile between defocus frames, consistent with the behavior seen in the simulated through-focal datasets. Finally, the experimental MEP blur is broader than the simulated value (22 Å) because a stronger regularization (0.5 vs 0.1 in simulation) was used for stable convergence, which necessarily suppresses high-frequency detail. The revised text (lines 207–224) now explicitly discusses geometry (α vs β), dose/SNR, and regularization as the three key contributors to the observed resolution:

“From isolated single-atom features in the experimental MEP reconstructions, we measured an axial blur of 40 ± 7 Å (Figure 4 and Supplementary Figure 5). Equivalent estimates were not possible for tf-ADF or tf-iDPC because finite-dose images are too noisy (consistent with limitations seen in simulations) and individual atoms likely move between defocus frames. Thus, MEP provides sufficient detail for quantitative 3D analysis, while tf-iDPC and tf-ADF could not reliably recover depth information under comparable conditions (Supplementary Table 3).

Two factors determine the practical resolution. First, the geometric limit is set by α for tf-ADF and tf-iDPC, and by the detector's outer collection angle β for MEP. In our experiments, our choice of β was constrained by the ~ 40 nm specimen thickness and the detector's finite pixel count, which limited the attainable collection angle without undersampling. Second, dose can also limit the achievable resolution: because each method accesses a different fraction of the scattered electrons (Figure 2a), not all have enough signal to reach their geometric limits. MEP collects nearly all scattered electrons ($\sim 99\%$), whereas tf-ADF uses $<1\%$, and tf-iDPC—despite the high SNR in the bright-field disk—cannot convert this advantage into reliable depth localization due to uncorrected multiple scattering. As shown in Figure 2g, tf-iDPC's axial profile is broadened to nearly the full sample thickness, reflecting its practical limitation. Additionally, the experimental MEP depth resolution is broader because a stronger regularization (0.5 vs. 0.1 in simulation) was used for stable convergence, which inherently suppresses high-frequency detail.”

14. Moreover, is the dose for a 2D image acquisition in both MEP and DPC/ADF is the same? ADF/DPC requires many through-focus slices which multiplies the dose requirements. Did the authors reduce the dose for each of the through-focus slices in DPC/ADF or kept it the same? It would be good to clearly establish at the beginning what are the fundamental data acquisition differences between DPC/ADF vs MEP, in terms of dose, data acquisition etc.

Response:

In the text, we report the total dose used for each imaging modality to obtain the 3D stacks, this means that for MEP this value is for the single 4D-STEM dataset, while for the tf-ADF/tf-iDPC it is the total dose for the whole series of the images, since here we are answering the question of, if we have X total electrons we can use to obtain a 3D image of the structure, how does each method do? The total doses used are stated in the caption of each comparison figure (Figure 2-3, Supplementary Figure 1-4). This was also stated in the Methods section, under Data Acquisition (lines 511-525):

“Data Acquisition

.... *A total dose of $0.5 - 2.5 \times 10^5$ e-/Å² was used for the 4D-STEM datasets.*

The DPC images were acquired on the Thermo Fisher Scientific Panther detector (8 sections) and the ADF images were acquired using the Thermo Fisher Scientific ADF detector, with *total through focal series dose of 1×10^5 e-/Å²* using an automated acquisition method outlined here⁷².”

In the experiments, the tf-ADF/tf-iDPC images were allocated a higher dose since at the same dose as MEP the images would simply be too noisy to interpret in any meaningful

way, and, in our experience, this was the maximum dose before substantial sample damage. We have made the wording more explicit in the paragraph in question to specify that the doses refer to a single MEP dataset and a full tf-ADF/tf-iDPC series (lines 204-206):

“MEP achieves a lateral information limit of 0.49 Å, compared to 0.66 Å for tf-iDPC and 0.83 Å for tf-ADF (Supplementary Figure 4), using only half the electron dose ($0.5 \times 10^5 \text{ e}^-/\text{Å}^2$ for the single MEP dataset versus $1 \times 10^5 \text{ e}^-/\text{Å}^2$ for the full tf-iDPC and tf-ADF image series).”

For the full experimental parameters, also see response to comment 12.

15. Line 196 - The following paragraph discusses strain, but it is not explained how they authors obtain the strain information. The resulting MEP reconstruction is simply the 3D structure of the sample. How is the strain recovered from the acquired data?

Response:

We thank the reviewer for the comment and clarify that the procedure for obtaining strain from the MEP reconstruction is described directly in the paragraph beginning at Line 210 (of the originally submitted manuscript, line 255 in revised manuscript) and detailed in the *Methods* section. Briefly, strain is quantified by tracking the position of silicon atomic columns in each depth slice using Atomap (see *Methods*, Supplementary Figure 7, and Supplementary Movies 1–3). The atomic positions are grouped into bilayers from the c-Si/a-SiO₂ interface inward, and Si–Si spacings are measured per bilayer to extract the strain profile, as shown in Figure 6.

16. Line 226 - "In both cases two fitting parameters are used...". Could the authors please elaborate what is actually being fitted? I would assume that the authors are fitting the extracted atoms and their positions using Atomap, but it is just a guess on my part.

Response:

We appreciate the reviewer’s attention to this point. As described in the *Methods* section (“Interface Roughness Quantification and Mobility Estimation”, line 618 onwards) and shown in Supplementary Figure 8, the fitting refers to the autocorrelation function of the measured interfacial roughness, not to the atomic positions themselves. The RMS amplitude (Δ) and correlation length (Λ) are obtained by fitting the autocorrelation envelope with exponential and Gaussian models. We have added a brief description to the main text for clarity (lines 272-274):

“... In both roughness models two fitting parameters are used: RMS amplitude Δ and correlation length Λ , extracted by fitting the autocorrelation function of the measured roughness with exponential or Gaussian models (see *Methods* and Supplementary Figure 8).”

17. Line 249 - "In the long-wavelength limit...". Could the authors briefly explain what is the wavelength referring to here and how it relates to the carrier mobility mentioned previously.

Response:

The "wavelength" here refers to the spatial wavelength of the charge carriers (or equivalently, the inverse of their momentum \mathbf{k}) relative to the characteristic length scale of interface roughness features. In the long-wavelength limit, the carrier wavelength is much larger than the roughness correlation length Λ , such that carriers experience the interface as an averaged potential variation. Under this approximation, standard interface-roughness scattering formalisms simplify to the mobility scaling of $\mu \propto 1/(\Delta^2\Lambda^2)$. We have clarified this point in the revised text by explicitly stating what the wavelength refers to and how this limit connects to carrier mobility (lines 296-301):

"These quantitative values of Δ and Λ can be linked to carrier mobility using established scattering formalisms. In the long-wavelength limit, **where the carrier wavelength is much larger than the characteristic interface roughness scale Λ , the carriers experience the interface as a smooth averaged potential, and** interface roughness scattering rates scale approximately as $1/(\Delta^2\Lambda^2)$ and reduce mobility as $\mu \propto 1/(\Delta^2\Lambda^2)^{68}$. At shorter wavelengths, the dependence is more complex and requires full evaluation of the roughness power spectrum."

18. Line 271 - "Importantly, for the information gained, MEP is relatively time- and resource-efficient: from sample preparation to a fully reconstructed 3D image takes only about two days, providing rapid atomic-scale structural feed". Could the authors expand a bit more on the time distribution for each step? E.g., how much slower is the sample preparation compared to imaging itself and the data reconstruction.

Response:

We appreciate the reviewer's interest in the practical timing of the workflow. These details have been added to the *Methods* section to clarify the typical time distribution (lines 553-561).

"Typical MEP workflow duration

The total measurement time for MEP experiments is comparable to standard high-quality TEM workflows. Focused ion beam (FIB) sample preparation for a <50 nm lamella typically requires half a day for an experienced user, followed by approximately half a day of microscope time (including alignment and tuning). Each 4D-STEM dataset (256 x 256 scan positions) is acquired in about 7 seconds on the EMPAD-G2 detector. Reconstruction typically takes from an hour to a day, depending on user familiarity, computational resources, degree of convergence needed for imaging vs quantification, and the algorithm used. These times are representative of our setup and will vary between laboratories."

19. Line 271 - The paragraph starting here seems to be related more to the "general discussion" and seems unrelated to the findings within the current section. Consider integrating into the discussion.

Response:

We agree with the reviewer that the final paragraph of this section functioned more as general discussion. This paragraph has been moved to the Discussion section (starting on line 332).

Minor comments

1. Line 37 - explain FET abbreviation (Field effect transistor).

Response:

We have added the full form field-effect transistor (FET) upon first mention in line 38:

"GAA transistors, with their nanosheet-based design, provide superior electrostatic control by completely surrounding the channel with the gate electrode, unlike planar **field-effect transistors (FETs)** or FinFETs, which lack full gate-channel coupling¹⁻⁵."

2. Line 45 - "X-ray ptychographic tomography" rather than just "X-ray ptychography".

Response:

Revised to "X-ray ptychographic tomography" (line 48):

"X-ray **ptychographic tomography** provides 3D imaging at deep sub-micron scales, ..."

3. Line 54 - "projection" to "violation of the projection approximation".

Response:

We have made this change (line 58).

4. Line 79 - if crystalline-Si is c-Si, please add the abbreviation in brackets, which appears later in the text.

Response:

Added the abbreviation "c-Si" after "crystalline Si" and "a-SiO₂" after amorphous-SiO₂ for consistency:

Line 85: "... Using MEP on early-stage GAA test structures from Interuniversity Microelectronics Centre (imec), we directly image stacking faults, pinholes, interface roughness, and strain relaxation in the **crystalline-Si (c-Si)** channel, capturing buried detrimental structural features well before electrical testing is possible."

Line 91: "... Comparing these findings to a conformal **amorphous-SiO₂ (a-SiO₂)** layer on epitaxial c-Si, we find that the strain relaxation is process-dependent and can serve as a metric for interface quality."

5. Line 81 - could the authors please refer here (and int his paragraph in general) to Figure 1 b such that it is clear what the text is referring to. Otherwise for people not familiar with the GAA structure it is unclear.

Response:

We have added a reference to Figure 1b in this paragraph to help orient readers with the GAA device structure (lines 89-91):

“In these GAA devices, **in the c-Si channel (labelled c-Si in Figure 1b)**, we measure the structural transition from strained interfacial silicon to bulk-like silicon to span over 2 nanometers of the 5-nanometer-wide channel (over 40%).”

6. Line 111 - replace "distortions in contrast" to "contrast distortions".

Response:

This entire paragraph has been revised to better explain the effects of non-linear probe-sample interactions and the original phrasing of "distortions in contrast" has been replaced by a more comprehensive description of the resulting artifacts, including axial elongation, contrast reversals, and positional inaccuracies (lines 121-130).

7. Line 132 - "This enables resolution beyond the diffraction limit imposed by the probe convergence semi-angle α ". The probe convergence does not impose a "diffraction limit", only the wave propagation physics imposes a "diffraction limit".

Response:

We thank the reviewer for this opportunity to clarify. The sentence has been revised (lines 161-165):

“ By capturing scattering to high angles (β , Figure 2a), these detectors extend the accessible spatial frequencies, enabling lateral and depth resolution **beyond the diffraction limit arising from the limited probe-forming-aperture semi-angle, α** . Under high-dose conditions, the ultimate resolution in MEP is therefore bounded by the detector’s maximum collection angle β ($> \alpha$) rather than α alone (Figure 2a)^{28,47-49}.”

8. Line 144, 223 - "S/TEM" and "(S)TEM". Please use a consistent naming convention.

Response:

We have standardized the terminology to use “STEM” when referring to specific experiments performed in scanning TEM mode, and “(S)TEM” only in general statements applicable to both TEM and STEM (e.g. when discussing sample preparation or general electron microscopy principles).

9. Line 149 - would be good to mention that "c" stands for crystalline and "a" for amorphous.

Response:

We have clarified in the text that “c” and “a” denote crystalline and amorphous, respectively (lines 175-176).

“Figure 3 summarizes the simulated 3D benchmark of a rough **crystalline-silicon (c-Si)/amorphous-hafnium oxide (a-SiO₂)/amorphous-hafnium oxide (a-HfO₂)** interface,”

10. Line 197 - "depth-resolving power" to "depth-resolving capabilities".

Response:

The sentence has been revised (line 242):

“Building on MEP’s demonstrated depth-resolving capabilities, we now apply it ”

11. Line 223 - "for the first time determining" to "determine"

Response:

Changed to “determine” (line 281).

12. Line 533 - "due to linear atomic number contrast, all sections of the structure are visible." what does this mean exactly?

Response:

We have clarified the caption to indicate that “linear atomic number contrast” means MEP allows both light (Si, O) and heavy (Hf) atoms to be simultaneously visible, making all layers of the structure apparent (lines 379-381):

“.... d MEP depth slices from the region in b-c (light and dark orange planes) showing both light (Si, O) and heavy (Hf) atoms, visible due to the linear atomic-number contrast.”

13. Line 623 - "3D rendering of GAA 2's surface", what is the "2's surface"?

Response:

The text has been clarified to indicate that “GAA-2” refers to the second of the two GAA regions analyzed in this study: GAA-1 and GAA-2 (lines 479-480):

“.... d 3D rendering of the second GAA (GAA-2), showing the full channel morphology.”

14. Figure 1 b - it is unclear what the text is pointing to exactly. Could the authors highlight and assign the layers better by e.g., a contour for each layer. Otherwise it is unclear.

Response:

We have revised Figure 1b to make the layer assignments clearer. Specifically, we added three arrows indicating the extents of the c-Si, a-SiO₂, and a-HfO₂ layers, and updated the labels to match these arrows. We also simplified the on-image text to avoid overcrowding. The caption has been revised to explicitly list each layer and describe the arrows (lines 371-375):

“.... **b** ADF image of a single gate-all-around structure with labeled components (crystalline-silicon (c-Si)/amorphous-hafnium oxide (a-SiO₂)/amorphous-hafnium oxide (a-HfO₂); thin black and white arrows indicate the extents of the Si channel and surrounding amorphous layers). Hafnium oxide intrusions into the silicon channel are visible, one of them highlighted with a red arrow.”

15. Figure 2 c - "Visualization of depth-resolved slices obtained by different methods". I only see three slices, which I assume are from MEP. Where are the other slices "obtained by different methods" which here should include also through focal series? Description seems misleading.

To clarify, Figure 2c (and panels 2a–d) are schematic illustrations rather than experimental data. Panel 2c is intended to conceptually show how depth-resolved slices can, in principle, be obtained using either MEP or through-focal (tf-ADF/tf-iDPC) approaches. We have revised the caption to state this explicitly and added arrows from panel 2d to 2c to emphasize that these slices are illustrative and not independent experimental reconstructions.

To improve completeness and better align with the manuscript’s revised flow, we have merged the original Figure 4 into the new Figure 2, so that experimental depth-resolved comparisons for all three methods (MEP, tf-ADF, tf-iDPC) now appear directly alongside the schematic panels. This provides the reader with both the conceptual illustration and the corresponding experimental realizations in a single figure, addressing the ambiguity noted by the reviewer. The updated figure is presented below (lines 384-409):

“Figure 2. Experimental configurations and depth-resolved imaging comparisons for MEP, tf-iDPC, and tf-ADF. a STEM geometry illustrating the converged electron probe (green, semi-angle α) raster-scanned (orange arrow) over an electron-transparent sample. Transmitted electrons are recorded using different detectors to generate distinct imaging modes: a high-angle annular dark-field (ADF) detector (blue), a quadrant detector for iDPC (purple), and a pixelated detector (EMPAD, black, 128×128 pixels) that records most scattered electrons up to the detector’s outer collection angle β . **b** Schematic of the multislice electron ptychography (MEP) reconstruction, where the probe is untangled from the sample potential to recover a stack of potential slices $V_1 \dots V_n$. **c** Conceptual illustration of depth-resolved slices reconstructed by different approaches, shown here using an atomic model of a GAA device. Green arrows from panel b and black arrows from panel d indicate that such depth resolved information can, in principle, be obtained from either MEP (single dataset) or through-focal (n scans of tf-ADF/tf-

iDPC) data. d Through-focal imaging scheme for tf-iDPC and tf-ADF, where a series of 2D images is recorded at different defocus values f_n and aligned to form a 3D representation of the sample convolved with the probe's point-spread function. Unlike MEP, these methods assume a weakly interacting probe and therefore neglect probe evolution from multiple scattering, leading to characteristic artifacts. **e–g** Depth-specific experimental comparison for the same GAA device region. **e** MEP reconstructed electrostatic potentials show an intact channel at 13 nm depth, a partially missing channel at 22 nm, and a distorted channel shape at 28 nm (yellow dashed outlines). Total dose: $0.5 \times 10^5 \text{ e}^-/\text{\AA}^2$. **f** tf-ADF images at corresponding defocus values (square-root contrast shown) obscure Si lattice contrast due to dose limitations and fail to recover the correct c-Si channel shape in depth. Total dose: $1 \times 10^5 \text{ e}^-/\text{\AA}^2$. **g** tf-iDPC images show the channel appearing intact at all depths, with poorly resolved lattice details and spurious depth information, missing the channel deformation. Total dose: $1 \times 10^5 \text{ e}^-/\text{\AA}^2$. Additional defect comparisons and depth slices are provided in Supplementary Figure 4.”

Reviewer #3 (Remarks to the Author):

Karapetyan et al describe a carefully written study using multislice electron ptychography (MEP) to study the nanoscale local structure of an advanced gate-all-around (GAA) nanosheet transistor. GAA transistor structures are central to next generation microelectronics. Characterizing the nanoscale structure of these nanodevices is a critical metrology challenge for the microelectronics industry. This report describes how MEP can provide nanoscale interface roughness and directly measure the local strain in the silicon channel of a single GAA transistor. They compare MEP to other electron microscopy modes and show how it improves what can be extracted from the microscope. Below I detail a couple of minor improvements that can be made to the manuscript. The manuscript will be of wide interest to the readership of Nature Communications.

Seeing this level of detail on a single device is amazing and will be of great interest to the semiconductor industry.

We thank the reviewer for their thoughtful and encouraging assessment of our work, and for recognizing the significance of applying Multislice Electron Ptychography (MEP) to Gate-All-Around (GAA) transistor metrology. We greatly appreciate the reviewer's positive comments on both the technical rigor and the broader relevance of this study to the semiconductor industry.

The authors should note that a chip has billions of transistors and that there is often variation between each individual device. That variation is quite important for product performance. The unaddressed point is statistical sampling. Measuring one out of a population of one billion is not statistically representative of the population. If they measured a second transistor it likely would be different from the first one. This makes the discussion in the manuscript about speeding up the measurement and reconstruction algorithm critical to the utility of the measurement so that it can be applied to many transistors to improve the statistics. I should note that for the purposes of the manuscript it is acceptable that they only report on one transistor device.

We thank the reviewer for the insightful comment regarding statistical sampling and device-to-device variability and the recognition that for this manuscript it was not necessary to measure multiple devices, although we did (Supplementary Figure 9 actually shows 3 device structures that have been fully analyzed – GAA-1, GAA-2 and a planar structure). Nevertheless, the editor asked us to comment further on this. We agree that statistical sampling across multiple devices is essential for capturing transistor-to-transistor variability, although we should note that this is not the typical workflow for microscopy in metrology or failure analysis today. Usually, variability is captured by electrical measurements from which bad devices are identified and individually targeted

for cross-section TEM by FIB liftout—a process we can supplement or replace straightforwardly with our approach.

Here, we showed that 3D atomic-scale imaging of a single device is now achievable with MEP and that such measurements are becoming practical: what once took weeks for reconstruction can now be completed within an hour (our record is 40 minutes) to a day. More importantly, once the microscope is aligned, a 3D data set for a single transistor takes 7 seconds to record and over a hundred devices can be imaged from a single TEM FIB section. With a little automation, all these devices could be imaged in about 30 minutes. The reconstructions can be processed in parallel on cluster, so the total reconstruction time still can be under hour. Even if we don't improve the algorithm (and we do expect to), Moore's law alone will halve the reconstruction time every 18 months from improved processor speeds alone. This progress makes large-scale statistical studies a realistic next step, and our results are a proof-of-concept that the method is ready for that transition.

Below, we provide detailed, point-by-point responses addressing all reviewer comments.

Comment #1:

It would be helpful to non-microscopists to include some additional diagrams in the figures to help one understand where in the device the particular view/slice comes from. The slices sometimes can be non-intuitive, especially with the asymmetric resolution. For example, on figure 1 it would be helpful to add marks on the 3D image in 1C highlighting where in z the slices shown in 1d come from. They should be color coded better. Making 1D the same green as 1C is a little confusing.

Response:

We thank the reviewer for this helpful suggestion. We have updated Figure 1 to improve clarity for readers who may be less familiar with slice orientation in 3D reconstructions. Specifically, we added light and dark orange markers in Figure 1c to indicate the z-positions of the slices shown in Figure 1d (also colored accordingly), and we revised the caption to describe these additions (lines 363-383):

“ **c** 3D MEP reconstruction showing interface roughness, a hafnium oxide intrusion (red arrow) and a step edge at the crystalline-silicon/amorphous-silicon oxide interface (green arrow). These buried features, important for device performance, while readily apparent with MEP, can be inaccessible or easily missed with conventional imaging methods. **d** MEP depth slices from the region in **b-c** (light and dark orange planes) showing both light (Si, O) and heavy (Hf) atoms, visible due to the linear atomic-number contrast. The hafnium oxide intrusion (top, **light orange slice**, $z = 10\text{nm}$, red arrow) is localized in depth and does not extend through the entire interface (bottom, **darker orange slice**, $z = 16\text{nm}$).”

Comment #2:

Extended data figure 4 - not clear from the caption what is different between the two images. The figures also appear to have different scalings in the two directions. The vertical dimension is 20 nm according to the arrows and the horizontal scale bar is 1 nm. The difference in scaling is about 3.67x.

Response:

The old Extended figure 4 showed two y-z depth slices through a planar device at 2 different locations in the x-y plane. We have updated the figure to add the xy area the section came from. To address reviewer 2's request for quantitative measurements of depth resolution, we have further updated this figure and moved it to the main section (now Figure 4, see below, with an extended version in Supplementary Figure 5). The difference in scaling reflects the difference between the lateral and depth resolutions and samplings, now highlighted in the caption (lines 434-454).

Figure 4. Experimental MEP depth sections from a planar c-Si/a-SiO₂/a-HfO₂ interface and corresponding depth blur estimation. a. Projected xy view of the MEP-reconstructed electrostatic potential of a planar c-Si/a-SiO₂/a-HfO₂ interface; the red dashed line marks the location of the depth slice in panel **b**. **b** This MEP depth section (with entrance surface on the left, and exit surface on the right) shows internal features of the amorphous HfO₂ layer, including nanoscale contrast variations consistent with short-range order or local density fluctuations. In contrast, the a-SiO₂ portion appears more homogeneous, consistent with its lower atomic number and more uniform amorphous structure. The characteristic triangular wedge geometry of the FIB-prepared TEM lamella is visible (outlined by dashed guidelines). A single atom located at the top surface (red arrow) is identified as redeposited from the TEM sample preparation. These observations demonstrate the sensitivity of MEP to both intrinsic and preparation-induced structural features, highlighting its utility for detailed 3D characterization of amorphous-crystalline interfaces. **Note that the voxel size is anisotropic in the depth direction.** **c** Local axial intensity profiles along isolated single atoms from this reconstruction (see Supplementary Figure 5 for slice locations; the red arrow in panel **b** corresponds to the third profile shown) together with Gaussian-plus-linear-background fits. The resulting full-

width-at-half-maximum (FWHM) values are listed below each profile. Across all six atoms, the fits yield depth blurs of 30–48 Å, corresponding to an average depth blur of 40 ± 7 Å (s.d.). Equivalent measurements cannot be performed for tf-ADF or tf-iDPC because the finite-dose images were too noisy and individual atoms often shifted between consecutive frames.

Comment #3:

The methods section does not describe the fabrication of the silicon, silicon oxide, and hafnium oxide blanket film.

Response:

We have added a description of the blanket film preparation to the Device Fabrication section of the Methods (lines 501-505):

“The planar interface (c-Si/a-SiO₂/a-HfO₂) blanket film sample used for comparison was fabricated separately. The Si portion of this sample corresponds to the starting Si wafer, while the SiO₂ layer is the native oxide present on the wafer surface, which serves as an effective nucleation layer for subsequent ALD deposition. The HfO₂ film was deposited at 300 °C using ALD with alternating HfCl₄ and H₂O pulses and N₂ purge steps between pulses.”

Comment #4

I recommend mentioning in the abstract that the GAA devices are fabricated by imec. With the TSMC coauthors and no imec coauthors it is easy to assume the devices were made by TSMC.

Response:

We appreciate the reviewer’s suggestion and have added a statement in the abstract to clarify that the prototype GAA devices were fabricated by imec (lines 22-24):

“By performing electron ptychography on prototype gate-all-around transistors fabricated by imec we uncover and quantify distortions and defects at the interface of the 3D gate oxide wrapped around the channel.”

Responses to Reviewer Comments:

We thank all reviewers for their thoughtful, constructive, and detailed feedback. We appreciate the reviewers' positive assessment of the manuscript and their recognition of the improvements made in clarity and accessibility. All remaining comments, primarily from Reviewer #3, have been carefully addressed through revisions to the manuscript and detailed point-by-point responses below. Our responses are shown in **green** and revised or newly added text in the manuscript is shown in **bold blue**. Line numbers referenced in our responses correspond to the revised clean manuscript. We have also clarified the figure captions (Figures 2 (lines 432-435) and 6 (lines 485-486)) to distinguish between shaded regions representing statistical variability (standard deviation) and widths defined by physical metrics such as full width at half maximum from fitted profiles.

REVIEWER COMMENTS

Reviewer #2 (Remarks to the Author):

The authors have addressed all of my comments and have made substantial improvements to the manuscript. I greatly appreciate the hard work of the authors and would be happy to see this work published. I have nothing to add or critique at this stage.

We thank the reviewer for their positive assessment of our work. We appreciate the reviewer's recognition of the substantial improvements made in the revision and their support for publication.

Reviewer #3 (Remarks to the Author):

The authors have provided an extremely thorough and careful response to the reviews. I thank them for that. The clarity of the manuscript has improved a lot. I think it now would be clear to both TEM people not familiar with advanced semiconductor devices and semiconductor people that are not TEM experts. There are a few minor issues that would further improve the manuscript.

We sincerely thank the reviewer for their careful reading of the revised manuscript and for their thoughtful and positive assessment and for the constructive suggestions, all of which we address in detail below and have incorporated into the manuscript to further improve clarity and precision.

Comment #1:

In the introduction, the authors distinguish between “light” elements and “heavy” elements and changes in the contrast for different imaging methods associated with atomic number. Later in the manuscript in the caption of figure 1 they state that light for GAA devices means Si and O and heavy is Hf. It would be valuable to say that in the main text where light and heavy are discussed in the introduction (page 4).

Response:

We thank the reviewer for this helpful suggestion. We have clarified the meaning of “light” and “heavy” elements in the context of GAA devices (lines 74-78):

“Here, we show how multislice electron ptychography (MEP) addresses the critical metrology gap in device characterization highlighted above, namely, the need for Ångström-scale resolution, three-dimensional imaging capabilities, and simultaneous sensitivity to both light and heavy atoms (**in the context of GAA devices, “light” refers to Si and O, while “heavy” refers to Hf**), thereby enabling direct measurement of atomic structure, strain, and interface roughness in 3D.”

Comment #2:

Paragraph starting on line 74, page 4. Somewhat redundant multiple uses of “light and heavy elements” and sub-Angstrom resolution (lines 75-77 and 81-83). “the need for Angstrom-scale resolution three-dimensional imaging capabilities, and simultaneous sensitivity to both light and heavy atoms” and “MEP has been established as a technique capable of sub-Angstrom in-plane resolution and nanometer-scale depth resolution, with sensitivity to both light and heavy elements.” They can combine those sentences to smooth the flow.

Response:

We thank the reviewer for this suggestion. We have revised the paragraph to eliminate redundancy by incorporating the discussion of sub-Ångström resolution and sensitivity to light and heavy elements, along with the corresponding citations, into the initial description of MEP (lines 74-84):

“Here, we show how multislice electron ptychography (MEP) addresses the critical metrology gap in device characterization highlighted above, namely, the need for Ångström-scale resolution, three-dimensional imaging capabilities, and simultaneous sensitivity to both light and heavy atoms²⁷⁻²⁹ (in the context of GAA devices, “light” refers to Si and O, while “heavy” refers to Hf), thereby enabling direct measurement of atomic structure, strain, and interface roughness in 3D. By scanning a convergent electron probe across the sample and collecting diffraction patterns from overlapping illuminated regions, MEP retrieves the phase shift introduced by the object’s atomically-resolved electrostatic potential, providing **nanometer-scale** depth information through post-processing of scanning diffraction data²⁶. ~~MEP has been established as a technique capable of sub-Ångström in-plane resolution and nanometer-scale depth resolution, with sensitivity to both light and heavy elements²⁷⁻²⁹.~~ We demonstrate that MEP enables high-fidelity three-dimensional imaging of GAA transistors, validate its accuracy on simulated structures and show experimentally that it outperforms conventional through-focal STEM in resolution and dose efficiency.”

Comment #3:

Page 9 (lines 195-199) – The transition between simulated and real data is not clear in the text. They do clearly state that the GAA is experimental, but it would be good to say when they change from silicon/silicon oxide/hafnium oxide to the Ta-filled pinhole sample that it is still simulated microscopy.

Response:

We thank the reviewer for pointing this out. We have revised the text to explicitly state that the Ta-filled pinhole example is based on a simulation study (lines 197-201):

“The improved depth fidelity of MEP translates directly to the ability to recover buried structures: **using a simulation study of a pMOS device model⁶¹**, Supplementary Figure 3 shows that MEP resolves an amorphous Ta-filled pinhole at 14 nm at the correct depth,

whereas the feature is indistinct in tf-iDPC and barely visible in tf-ADF (with the silicon channel completely obscured), despite the latter using more than twice the dose.”

Comment #4:

Figure 4c – According to the caption, the 3rd slice in 4c corresponds to the single atom marked by an arrow in 4b. The other slices correspond to single atoms shown in the supplemental. I recommend they put a red box around the slice in 4c that corresponds to the red arrow in 4b.

Response:

We have updated Fig. 4c to explicitly highlight the axial profile corresponding to the single atom marked by the red arrow in Fig. 4b by enclosing it in a red box. The caption has been revised accordingly to clarify the correspondence between panels (lines 443, 458):

“... the red arrow in panel b corresponds to the third profile shown, **boxed in red**). ...”

Comment #5:

On page 5 (lines 95-100) and page 15 (lines 328-329) the manuscript mentions electrical testing would require months of fabrication to produce a testable device. Full production

chip take months to produce, but during process development fabs make short loop wafers with a small number of metal layers to allow function testing after a much shorter time (days to weeks). This is critical for development of new device structures/processes since at that time they have no idea how to relate images of the devices to end performance. The current wording is a little misleading since no fab would take multiple months to be able to etest structures during new device development. The wording seems to merge process monitoring during production where an early metrology step could save a lot of time/cost and metrology during process development where it helps them know what they made and relate structure to electrical function. Separate from etest, 3D atomic scale resolution would speed up the development cycle by speeding up learning.

Response:

We thank the reviewer for prompting us to clarify the language regarding electrical testing during process development. To remove any potential ambiguity, we have revised the manuscript to more explicitly distinguish between short-loop electrical testing and the full fabrication flow, and to clarify the role of MEP during process development. Specifically, we revised the text to emphasize that MEP performed on dedicated witness wafers accelerates the learning cycle during process development by enabling direct correlation between process changes and buried structural outcomes, complementary to short-loop electrical testing (lines 94-102):

"Formation of the critical GAA structure occurs early in the overall CMOS fabrication process, which comprises roughly a thousand steps and typically takes 3-4 months to complete from start to finish. Early structural feedback from MEP performed on dedicated witness wafers (test wafers fabricated alongside production wafers for process monitoring, including by destructive analysis techniques) can provide rapid insight into structural defects and interface quality early in **the** fabrication flow. **This will accelerate the learning cycle during process development by enabling direct correlation between process changes, structural outcomes, and complementary short-loop electrical testing**, reducing costly iterations."

We also revised lines 328–329 to clarify that MEP can be applied after any fabrication step during process development to provide early atomic-scale structural feedback, without implying that any electrical testing requires completion of the full fabrication flow. The revised text now reads (lines 329-336):

"MEP can be applied after any fabrication step during process development, providing early three-dimensional atomic-scale structural feedback well before completion of the full fabrication flow. This information complements short-loop electrical testing, which primarily probes aggregate electrical parameters such as current, capacitance, and resistance, by instead directly characterizing buried structural variations and process-dependent interfacial disorder that are not accessible through electrical measurements alone or through conventional

projection-based TEM imaging. By providing this complementary structural information, MEP enables faster learning and refinement of fabrication processes.”

In addition, we note that while short-loop electrical testing is an essential part of process development, the additional processing steps required to enable electrical access (including metallization and associated thermal or annealing treatments) can themselves modify the atomic-scale structure formed at earlier stages of fabrication, that are the steps of interest for process development. As a result, structural characterization performed immediately after formation of the feature of interest, prior to subsequent thermal steps, provides complementary and, in some cases, uniquely informative insight into the intrinsic structural outcome of a given process step. This further highlights the value of MEP as an early-stage, step-resolved metrology tool that accelerates learning during process development rather than replacing electrical validation.

Comment #6:

How does computation time scale with the size of the measured volume? Since the 4D STEM measurement is pretty fast, end users would be interested in trying to get a larger reconstruction volume.

Response:

We thank the reviewer for this question. The reconstruction time in multislice electron ptychography depends on several factors, including the number of probe positions, detector pixels, number of reconstructed depth slices, and number of reconstruction iterations, and therefore does not follow a single simple scaling law with volume. In practice, for the datasets considered here, reconstruction time scales approximately with the total number of measured probe positions and the number of reconstructed depth-slices and can be effectively mitigated through the reconstruction of multiple datasets in parallel (this is the workflow used in our MEP reconstructions), i.e. it uses more compute resources, but does not take any longer. We have revised the Methods section to clarify that while larger reconstruction volumes increase computational cost, the reconstruction workflow is still practical for device-scale volumes (lines 565-577):

“The total measurement time for MEP experiments is comparable to standard high-quality TEM workflows. Focused ion beam (FIB) sample preparation for a <50 nm lamella typically requires half a day for an experienced user, followed by approximately half a day of microscope time (including alignment and tuning). Each 4D-STEM dataset (256 x 256 scan positions) is acquired in about 7 seconds on the EMPAD-G2 detector. Reconstruction typically takes from an hour to a day, depending on user familiarity, computational resources, degree of convergence needed for imaging vs quantification, and the algorithm used. These times are representative of our setup and will vary between laboratories.

Reconstruction time scales roughly linearly (with a small offset for setup) with the number of probe positions (i.e. area) and with the number of reconstructed depth slices⁶⁰; however, the iterative reconstruction is efficiently accelerated on GPUs and can be distributed across compute nodes. Thus, reconstructing a large area can be done in a similar time, provided more compute resources are available, enabling practical times for larger volumes and multi-device studies.”

Comment #7:

Typo (page 8, line 176) – “rough crystalline-silicon (c-Si)/amorphous hafnium oxide (a-SiO₂)/amorphous-hafnium oxide (a-HfO₂) interface.” The first hafnium oxide should be silicon oxide.

Response:

We thank the reviewer for catching this typo. The text has been corrected (lines 177-178):

“... Figure 3 summarizes the simulated 3D benchmark of a rough crystalline-silicon (c-Si)/amorphous-silicon oxide (a-SiO₂)/amorphous-hafnium oxide (a-HfO₂) interface, ...”

Comment #8:

The author’s general response to reviewer #3 about measurement statistics appears to mix up two different uses for 3D atomic scale resolution imaging – failure/fault analysis and development testing/feedback. Both are important. It is very difficult to make what you can’t measure, so fast data on process changes speed up the learning loop. In the case of development testing/feedback statistics is very important because they aren’t trying to look at a specific transistor to figure out why it didn’t work. They are trying to learn what changed in the transistors for a processing variation. None of the review response discussion about failure analysis was in the manuscript so the comment is only to correct the review response.

Response:

We thank the reviewer for this clarification and agree with the distinction between failure/fault analysis and development testing or process feedback. Our intent in the manuscript is to emphasize the latter, namely, the use of three-dimensional atomic-scale imaging to accelerate learning during process development by enabling atomic-scale 3D insight into structural changes arising from process variations, free from common projective S/TEM imaging artifacts. We agree that this context is distinct from single-device failure analysis, and we confirm that the manuscript text is framed around development feedback and learning rather than failure analysis. No changes to the manuscript were required.